# Groundwater fauna in an urban area: natural or affected?

Fabien Koch[1], Kathrin Menberg[1], Svenja Schweikert[1], Cornelia Spengler[2], Hans Jürgen Hahn[2], Philipp Blum[1]

[1]Institute of Applied Geosciences (AGW), Karlsruhe Institute of Technology (KIT), Kaiserstraße 12, 76131 Karlsruhe, Germany
[2]Faculty of Nature and Environmental Sciences (Working Group: Groundwater Ecology), University Koblenz-Landau, Im Fort 7, 76829 Landau, Germany

*Correspondence to*: Fabien Koch (fabien.koch@kit.edu)

**Abstract.** In Germany 70 % of the drinking water demand is met by groundwater, whose quality is the product of multiple physical-chemical and biological processes. As healthy groundwater ecosystems help to provide clean drinking water, it is necessary to assess their ecological conditions. This is particularly true for densely populated, urban areas, where faunistic groundwater investigations are still scarce. The aim of this study is therefore to provide a first assessment of the groundwater fauna in an urban area. Thus, we examine the ecological status of an anthropogenically influenced aquifer by analysing fauna in 39 groundwater monitoring wells in the city of Karlsruhe (Germany). For classification, we apply the groundwater ecosystem status index (GESI), in which a threshold of more than 70 % of Crustaceans and of less than 20 % of Oligochaetes serves as an indication for very good and good ecological conditions. Our study reveals that only 35 % of the wells in the residential, commercial and industrial areas, and 50% of wells in the forested area fulfil these criteria. However, the study did not find clear spatial patterns with respect to land use and other anthropogenic impacts, in particular groundwater temperature. Nevertheless, there are noticeable differences in the spatial distribution of species in combination with abiotic groundwater characteristics in groundwater of the different areas of the city, which indicate that a more comprehensive assessment is required to evaluate the groundwater ecological status in more detail. In particular, more indicators, such as groundwater temperature, indicator species, delineation of site-specific characteristics and natural reference conditions should be considered.

## 1. Introduction

In Germany 70 % of the drinking water demand is met by groundwater, whose quality is the product of multiple physical-chemical and biological processes (German Environment Agency, 2018). Groundwater ecosystems are responsible for several services that help to provide clean drinking water, which is a vital resource for humanity (Griebler and Avramov, 2015). Bacteria and fauna also play an important role in the biological self-purification of groundwater by the retention of organic matter, natural attenuation of pollutants, storing and buffering of nutrients as well as the elimination of pathogens. Organic matter and pollutants can be degraded and converted to biomass or bound by microbial activity. Protozoa and higher organisms can graze resulting biofilms, loosen the substrate and therefore stimulate biological self-purification (Hancock et al., 2005; Boulton et al., 2008; Griebler and Avramov, 2015).

Healthy groundwater ecosystems can provide clean drinking water, however, they are sensitive to external influences such as chemical and thermal disturbances. The latter drives hydro-geochemical and biological processes in groundwater systems which are typically isothermal (Brielmann et al., 2009; 2011). Groundwater fauna mainly consists of stygobiont species which spend their entire life in groundwater and are adjusted to this habitat (Hahn, 2006). Hence, in Central Europe they are assumed to be cold stenotherm which means that they prefer cold temperatures. A variability in temperature tolerance among groundwater faunal groups and species is reported in various studies, which explains why the use of individual temperature thresholds is more useful to capture different preferences. According to Spengler (2017) faunal diversity is generally declining at a temperature above 14 °C. Various authors reported species specific temperature preferences between 8 and 16 °C (for individuals of the species *Niphargus inopinatus* and *Proasselus cavaticus* (Brielmann et al., 2009, 2011)) and a specific temperature threshold of up to 19 °C (for *Parastenocaris phyllura* (Glatzel, 1990)). Above these thresholds the mortality of individuals raises until groundwater fauna is almost absent, for example at 22 °C in the study of Foulquier et al. (2011). However, temperature sensitivity is not only an issue at species level but also for the communities as a whole. Spengler (2017) reported 12 °C to be a temperature threshold value indicated by a shift in community structure for faunal communities of groundwater of the Upper Rhine Valley.

Nevertheless, in German and European legislation, as in many countries globally, groundwater is not yet recognized as a habitat which is worthy of protection and there is no common understanding on the best practice of assessing the ecological status of groundwater (Hahn et al., 2018; Spengler and Hahn, 2018). The assessment of surface water is typically based on biological, physical-chemical and supported by hydro-morphological criteria (European Water Framework Directive and German legislation article 5 of the 'Regulation on the Protection of Surface Water'). While groundwater quality is mostly assessed by physical-chemical and quantitative criteria, very few quantifiable ecological criteria are available for the assessment of the health of groundwater ecosystems. The availability of ecological criteria can only be increased by conducting a large number of studies dealing with the analyses of groundwater ecosystem health by investigating groundwater fauna. Results from previous faunistic groundwater analyses are contained in a Germany-wide data record (Hahn, 2005; Berkhoff, 2010; Stein et al., 2012; Gutjahr, 2013; Spengler, 2017; Spengler and Hahn, 2018). The study by Hahn and Fuchs (2009)

focuses on defining stygoregions based on different hydrogeological units located in Baden-Württemberg, Germany. They conclude that the observed patterns of groundwater communities reflect a high spatial and temporal heterogeneity of aquifer

types with respect to habitat structure, food and oxygen supply. Although there are various studies on this topic (e.g. Gibert and Deharveng, 2002; Malard et al., 2002; Deharveng et al., 2009; Dole-Olivier et al., 2009b) stygobiotic biodiversity is still likely to be underestimated.

Regional investigations on the spatial variation of groundwater fauna, i.e. stygobiont occurrences, and corresponding environmental parameters, such as geological site characteristics and altitude, are rare (Dole-Olivier et al., 2009a; Gibert et al.,

2009). An approach to elucidate groundwater biodiversity patterns in six European regions was conducted in the PASCALIS project (Protocol for the Assessment and Conservation of Aquatic Life In the Subsurface) (Gibert et al., 2009), which aimed at mapping biodiversity and endemism patterns (Deharveng et al., 2009) and shows that regional processes, such as hydrological connectivity in a specific habitat (e.g. river floodplains as in Ward and Tockner, 2001) have a much stronger influence on species composition than local habitat features such as permeability and saturation. Within a region,

hydrogeology, altitude, palaeographical factors and human activities can interact in complex ways to produce dissimilar patterns of species compositions and diversity (Gibert et al., 2009). The PASCALIS sampling protocol recommends selecting hydro-geographic basins that are not strongly affected by human activities such as groundwater pollutions (Malard et al., 2002), and does not biogeographically classify a groundwater system (Stein et al., 2012).

In urban areas, anthropogenic impacts such as a dense building development, underground car parks, open geothermal systems

and injections of thermal wastewater from industry result in local thermal alteration of groundwater up to several degrees (e.g. Taylor and Stefan, 2009; Zhu et al., 2011; Menberg et al., 2013b; Tissen et al., 2019). According to Brielmann et al. (2011) annual temperature fluctuations in aquifers caused by shallow geothermal energy systems, range between 4 °C in winter and $\leq 20$ °C in summer. In 2000, the European Union (EU) (Water Framework Directive) defined the release of heat in the groundwater as a pollution, whereas the cooling of the groundwater is not mentioned. Until now, there are scientifically derived

threshold values for groundwater temperature in the case of thermal (heat) pollution published, but none of these have been implemented in official regulations or water law (Hähnlein et al., 2010, 2013; Blum et al., 2021). This results in a tension between conservation, exploitation and thermal use of groundwater. However, as seen in an aquifer ecosystem downstream from an industrial facility in Freising (Germany), where groundwater is used for cooling resulting in a warm thermal plume, no relation between faunal abundance and groundwater temperature could be identified (Brielmann et al., 2009). Investigation

of hydro-geochemical parameters, microbial activities, bacterial communities and groundwater faunal assemblages indicates that bacterial diversity increased with temperature, while faunal diversity decreased with temperature (Brielmann et al., 2009). Similar results are provided by Griebler et al. (2016), where potential impacts of geothermal energy use and storage of heat on groundwater are investigated. Temperature changes in groundwater correspond with changes in groundwater chemistry, biodiversity, community composition, microbial processes and function of the ecosystem. How exactly groundwater

communities react to changes in temperature and concentration of nutrients, dissolved organic carbons and oxygen, is not yet fully understood (Brielmann et al., 2009, 2011; Spengler, 2017; Sánchez et al., 2020).

Several approaches exist that allow a local assessment of the ecological state of groundwater based on different faunistic, hydro-chemical and physical parameters. Korbel and Hose (2011, 2017) introduced the Groundwater Health Index (GHI), which is a tiered framework for assessing the health of groundwater ecosystems. Here, both biotic and abiotic attributes of groundwater ecosystems are used as benchmarks for ecosystem health. Their study shows that ecosystem health benchmarks are probably more associated with aquifer typology than being applicable for local areas. This index is applied and tested by Di Lorenzo et al. (2020) in unconsolidated aquifers in Italy located in nitrate vulnerable zones. They refined the index (wGHI$^N$) and demonstrated its applicability on shallow and deep aquifers and also revealed that this new index is limited due to low correlations between the indicators. Commissioned by the Federal Environmental Agency of Germany (Umweltbundesamt, UBA), Griebler et al. (2014) developed a concept for an ecologically based assessment scheme for groundwater ecosystems, which builds on the assessment of Korbel and Hose (2011, 2017). This two-step scheme characterizes groundwater on two different levels by using the most important physico-chemical parameters, such as content of dissolved oxygen, as well as microbiological and faunistic characteristics such as amount of Oligochaetes and Crustaceans, and comparing these to reference values for natural, undisturbed and ecologically intact groundwater ecosystems (Griebler et al., 2014).

Furthermore, the Groundwater-Fauna-Index (GFI), introduced by Hahn (2006), quantifies the relevant ecological conditions in the groundwater as a result of hydrological exchange between surface and groundwater. It incorporates ecologically important groundwater parameters such as relative amount of detritus, variation of groundwater temperature and concentration of dissolved oxygen (Hahn, 2006). Gutjahr et al. (2014) used the GFI as part of a proposal for a groundwater habitat classification on a local scale, which introduce five types of faunistic habitats as a result of surface water influence, content of dissolved oxygen and amount of organic matter. Moreover, in the study of Berkhoff (2010) the GFI was used to examine the impact of the surface water influence on groundwater with the aim to develop a faunistic monitoring concept for hydrological exchange processes in the surrounding river bank filtration plants. Spengler and Hahn (2018) argued for the definition of a regional and ecological temperature threshold and an ecology based assessment of thermal stress in groundwater.

The objective of this study is to investigate specifically the groundwater fauna beneath residential, commercial and industrial, i.e. urban areas in comparison to a forested area outside the built-up area of Karlsruhe to determine whether land use has an impact on groundwater faunal communities. Hence, in 39 groundwater monitoring wells in Karlsruhe, Germany, the groundwater fauna is sampled, groundwater temperatures measured and chemical properties are analysed. In our study the classification scheme developed by Griebler et al. (2014) is applied. The wells are characterized regarding the state of their ecosystem. Hence, we finally aim to distinguish areas with natural groundwater ecology from anthropogenically disturbed areas.

## 2. Material and methods

### 2.1 Study site

The study is performed in Karlsruhe, a city in the Upper Rhine Valley in south-western Germany. The urban region covers an area of 173 km$^2$ and has about 310,000 inhabitants (Amt für Stadtentwicklung - Statistikstelle, 2018). The Cenozoic continental rift valley is filled with Tertiary and Quaternary sediments, which are dominated by sands and gravels with minor contents of silt, clay and stones (Geyer et al., 2011). Sporadic layers with lower permeabilities lead to a separation of up to three aquifer levels (Wirsing and Luz, 2007). The upper aquifer is unconfined with a water table between 2 and 10 m below the ground. The flow direction is northwest of the Rhine River with groundwater flow velocities ranging between 0.5 and 1.5 m/d (Technologiezentrum Wasser, 2018).

Based on the land use plan of Karlsruhe, about 20 % of the area (i.e. urban area, city centre, neighbouring districts, as well as parts of the Hardtwald forest and several outskirts) is covered by buildings. The rest is vegetation (~ 56 %) and artificial surface covers (~ 24 %), showing the complexity and heterogeneity of the urban environment. According to Benz et al. (2016), the annual mean groundwater temperature (GWT) in Karlsruhe in the years 2011 and 2012 was 13.0 ± 1.0 °C. Distinct temperature hotspots occur mainly below the city centre, where building densities are highest. In the north-western part of Karlsruhe, the increase of GWT was about 3 K warmer than the annual mean land surface temperature (LST), which is mainly caused by several groundwater reinjections of thermal wastewater (Benz et al., 2016).

In general, groundwater in the region of Karlsruhe is of good quality and the local drinking water supplier (Stadtwerke Karlsruhe) only needs to remove oxidised iron and manganese from the pumped groundwater. However, two main contaminations which affect groundwater quality are known in the urban area (Stadt Karlsruhe, 2006). A contaminant plume, which contains a polycyclic aromatic hydrocarbons concentration of up to 500 µg/l, of 200 m length over the entire aquifer thickness is located at a former gas plant in the east of Karlsruhe (Figure S1b) (Kühlers et al., 2012). Moreover, three parallel contamination plumes of 2.5 km length each, can be found in the southeast of Karlsruhe (Figure S1b), where highly volatile chlorinated hydrocarbons (7 µg/l - 26 µg/l) and their degradation products were detected (Wickert et al., 2006).

### 2.2 Material and sampling

From 2011 to 2014, samplings of groundwater parameters and fauna were performed in 39 groundwater monitoring wells in the city area of Karlsruhe, of which eight wells are in the forested area and 31 in the residential, commercial and industrial areas (urban area). At the beginning of each sampling process, temperature and electrical conductivity were measured with an electric contact gauge (Type 120-LTC, Hydrotechnik) at a depth interval of 1 m. Using a bailer (Aqua Sampler, Cole-Parmer), water from the bottom of the groundwater monitoring wells was sampled and the pH value (Multiline Type 3430; WTW GmbH, Weilheim Germany) as well as the contents of dissolved oxygen (Multiline Type 3430; WTW GmbH, Weilheim Germany), iron, nitrate ($NO_3^-$) and phosphate ($PO_4^{3-}$) (RQflex® plus 10 Reflectoquant®; Merck Millipore KGaG, Darmstadt Germany) were measured.

In accordance with the suggestion made by Hahn and Gutjahr (2014), several integrative samplings (i.e. repeated samples taken over a period of time) were conducted to capture an ecological representation of groundwater fauna which reflects the occurring species at a community level. Every well is sampled at least three times. From 2011-2012, 22 measurement wells (mainly in the Hardtwald and the North-West of Karlsruhe) were sampled six times at a minimum interval of two months. In 2014, 17 measurement wells, mainly located in the south/inner city, were sampled three times (see Table S2). As the aim of this study is to provide a first-tier screening of the groundwater ecological status, we sampled the fauna in the monitoring wells in accordance with the sampling manual of the European PASCALIS Project (Malard et al., 2002) and the procedure described by Hahn and Fuchs (2009), using a modified Cvetkov net.

Furthermore, the relative amount of sediment as an indication of the nutrient availability and the cavity system was measured. Before the fauna sample from the net sampler was passed over a sieve with a mesh size of 74 µm, the sediment is separated and classified in different categories (sand, fine sand, ochre, detritus, silt). It should be noted that the detritus content is not recorded quantitatively but on the basis of estimated frequency classes. The estimation of the relative amounts of sediment per sample is based on Table S1 in the supplement.

Mann-Whitney-tests (U-tests) were applied to detect potential impacts of groundwater characteristics (physical-chemical parameters), geology and well design on the groundwater quality as well as on groundwater fauna. Samples were regarded as significantly different if the $p$-value was $< 5.0 \times 10^{-2}$.

To better understand large-scale relationships as well as fine structures of high-dimensional biological data, the PHATE (potential of heat diffusion for affinity-based transition embedding) analysis introduced by Moon et al. (2019) (https://github.com/KrishnaswamyLab/PHATE) was used. This dimensionality-reduction method generates a low-dimensional embedding specific for visualization, which provides an accurate, denoised representation of both local and global structures of a dataset without imposing strong assumptions on the structure of the data. The PHATE algorithm computes the pairwise distances from the data matrix and transforms the distances to affinities to encode local information by applying a kernel function, which is developed to Euclidian distances. By using diffusion processes, global relationships are learned and encoded using the potential distance. Finally, the potential distance information is embedded into low dimensions for visualization by using metric Multi-Dimensional-Scaling (MDS) (Moon et al., 2019). Objects that are close to each other in the final graph therefore have similar characteristics.

Crustaceans, especially Amphipods and Copepods represent the majority of groundwater fauna. The identification keys from the following studies were used to identify the different groups in the samples: Einsle (1993), Janetzka et al. (1996), Meisch (2000), Schellenberg (1942) and Schminke et al. (2007). The sampled fauna for this study can be assigned to the subphylum *Crustacea* and four other subordinate taxa (Table 1).

**Table 1: Overview of the sampled fauna, divided into the subphylum *Crustacea* and other subordinate taxa.**

| Subphylum: *Crustacea* | Size [mm] | Habitats | Species number |
|---|---|---|---|
| Order: *Cyclopoida*  | 0.4 - 0.7[1] | Fresh and marine water, groundwater[1] | 298 species and subspecies worldwide[2], 8 stygobiotic species in Germany[3] |
| Order: *Harpacticoida*  | < 0.5[4] | Marine, freshwater, semi-terrestrial environments and groundwater[5] | 599 (sub-)species worldwide[2], 20 stygobiotic species in Germany[3], 17 stygophile* & stygobiotic species in Baden-Württemberg[6] |
| Genus: *Parastenocaris* | 0.3 - 0.5[1] | Tertiary relict living in cavity rooms of streams, in groundwater and moss[1] | 206 (sub-)species worldwide[2] (16 stygophile & stygobiotic species in Baden-Württemberg[1]) |
| Order: *Bathynellacea*  | 0.5 - 5.4[7] | Cavity systems[7] and in groundwater[8] (foreign tropical origin)[9] | Exclusively 160 stygobiotic species worldwide[9], 8 species in Germany[3] |
| Order: *Amphipoda*  | 0.5 – 30[1] | Sea, fresh water[1] and in healthy groundwater ecosystems (important ecosystem service providers[10] & biodiversity indicators in Europe[11]) | 321 stygophile & stygobiotic species in Europe[12], 24 stygobiotic species in Germany[3] |

| Other subordinate taxa | Size [mm] | Habitats | Species number |
|---|---|---|---|
| Subclass: *Oligochaeta*  | < 1 – 3[13] | Colonise every habitat, groundwater[13] | 100 species worldwide[14] and 27 stygobiotic species in Europe[13] |
| Phylum: *Nematoda*  | 1 – 3[9] | Colonise every habitat[9], can live under unfavourable conditions[15] | 20,000 species worldwide[16], 60 stygobiotic species in Europe, 6 species in Germany[3] |
| Class: *Turbellaria*  | 0.4 – 5[17] | Sea, brackish and fresh water and groundwater[17] | 3,400 species worldwide[17], 7 stygobiotic species in Germany[3] |
| Subclass: *Acari*  | a few mm[9] | Colonize every habitat, also groundwater, have high demands on water quality[9] | < 5,000 water mite species wordlwide[18], 10 stygobiotic species in Germany[3] |

[1] Fuchs et al. (2006)
[2] Galassi (2001)
[3] Zenker et al. (2020)
[4] Hahn (1996)
[5] Galassi et al. (2009)
[6] Fuchs (2007)
[7] Sauermost and Freudig (1999a)
[8] Camacho (2006)
[9] Hunkeler et al. (2006)
[10] Boulton et al. (2008)
[11] Stoch et al. (2009)
[12] Botosaneanu (1986)
[13] Sauermost and Freudig (1999b)
[14] Batzer and Boix (2016)
[15] Hahn et al. (2013)
[16] Eckert et al. (2008)
[17] Sauermost and Freudig (1999c)
[18] di Sabatino et al. (2000)

*Stygophile organisms are found primarily in surface water, but they can survive in shallow groundwater for a while (Preuß and Schminke, 2004).

### 2.3 Classification scheme by Griebler et al. (2014)

Commissioned by the Federal Environmental Agency of Germany (UBA), Griebler et al. (2014) developed a two-step ecologically based classification scheme for characterization of groundwater ecosystems and also defined spatially dependent reference values of ecologically intact groundwater ecosystems. In order to enable a statement about the exposure of the groundwater at a specific site, biotic and abiotic parameters, which are determined and compared with reference values, are used to distinguish locations with very good or good ecological conditions or locations which fail these criteria, i.e. affected areas (Figure 1). If an ecological assessment of groundwater ecosystems, which is based on the groundwater fauna analysis, takes place, some faunistic criteria must be considered. Invertebrates avoid habitats that are ochred or have a low content of dissolved oxygen. Thus, unstressed or natural habitats are defined as areas with a content of dissolved oxygen > 1.0 mg/l, that are not ochred and have an existing fauna, i.e. an amount of > 50 % of Stygobites, of > 70 % of Crustaceans and of < 20 % of Oligochaetes (Figure 1). This allows a qualitative interpretation of the ecological condition of the groundwater system. If the results indicate affected ecological conditions, i.e. one or more biological/ecological indicators are out of the reference range, an assessment according to the Level 2 scheme is necessary. This requires a determination of reference values at local reference locations which are protected and have a weak surface influence as well as a subsequent comparison of these values with measured data. As our aim is a first-tier screening of an urban area, we only apply Level 1 in our study.

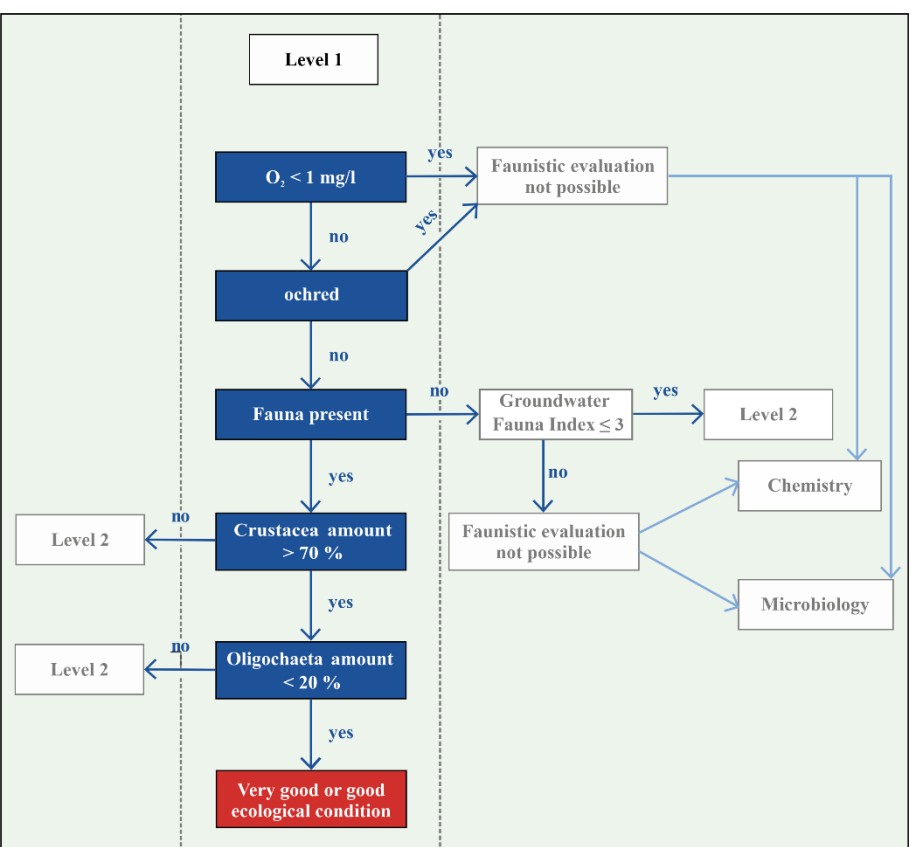

**Figure 1: Classification scheme by Griebler et al. (2014) according to Level 1 for groundwater ecosystems on the basis of groundwater fauna (modified after Griebler et al. (2014)).**

## 3. Results and discussion

### 3.1 Physical and chemical parameters

First, the groundwater conditions in the study site are evaluated by their physical-chemical characteristics. The following values are average values of the individual samplings from each monitoring well. In order to allow a spatially differentiated assessment, the study site (city area of Karlsruhe) is classified in different zones based on land use types provided by the European seamless vector database of the CORINE Land Cover (CLC) inventory (GISAT, 2016). Based on this data the city area is subdivided into:(1) Forested area (forest; local name: Hardtwald) and (2) industrial, commercial and residential areas (urban area) (Figure 2a). For simplification, the phrases 'forest' and 'urban area' are used in the following. A more detailed subdivision in the urban area did not appear reasonable due to the heterogeneous structure.

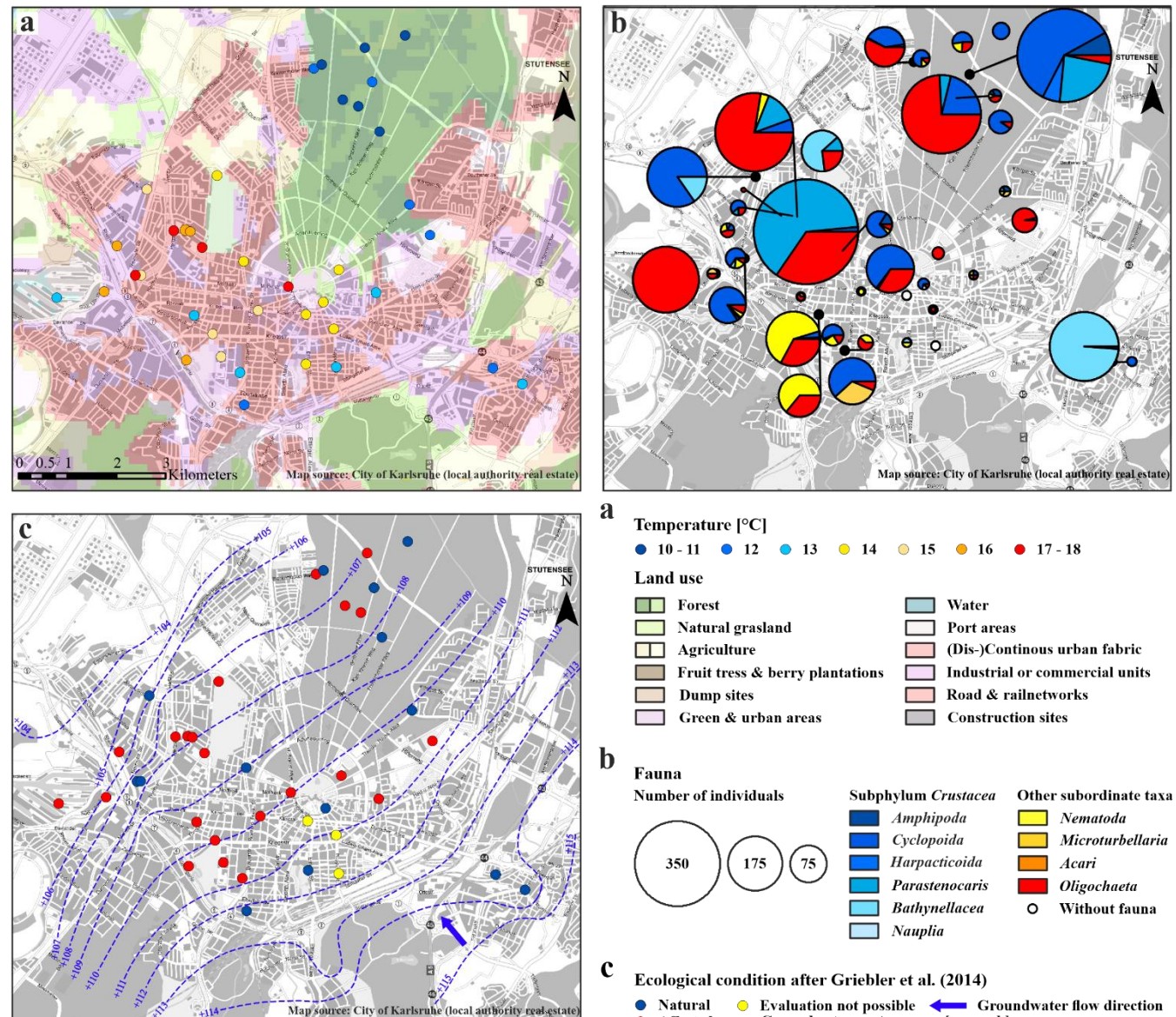

**Figure 2: Overview map city area of Karlsruhe: (a) land use plan (GISAT, 2016) and average groundwater temperature of the multiple measurements [°C] at the bottom of the monitoring wells; (b) detailed groundwater fauna: colours of the circles show the different taxa in the sample [%], the size indicates the number of individuals; (c) faunistic evaluation after Griebler et al. (2014) and groundwater contour map in metres above sea level (modified after the local authority real estate of Karlsruhe).**

As expected, measured GWT at the bottom of the wells in 8.5 to 39.0 m depth, are mainly constant over the repeated measurements. The lowest GWT ranging between 10.5 and 10.9 °C were measured in the eight wells of the forested area (Table S2). In contrast, the highest average GWT with 17.5 °C was measured in a well near the city hospital (T113) (Figure 2a). The mean value of all wells is 13.5 ± 2.1 °C, which is similar to the results from Benz et al. (2015) with 13.0 ± 1.0 °C. According to Benz et al. (2017), annual shallow GWT vary between 6 and 16 °C in the area of Karlsruhe, which is in line with

the temperatures measured during fauna sampling (Figure 3a). For the urban area in the north-western part of the city, Figure 2a shows a clear warming trend, which was also observed by Menberg et al. (2013a,b). The increased GWT in this area can be traced back to effects of urban infrastructures and industries, which use groundwater for cooling purposes.

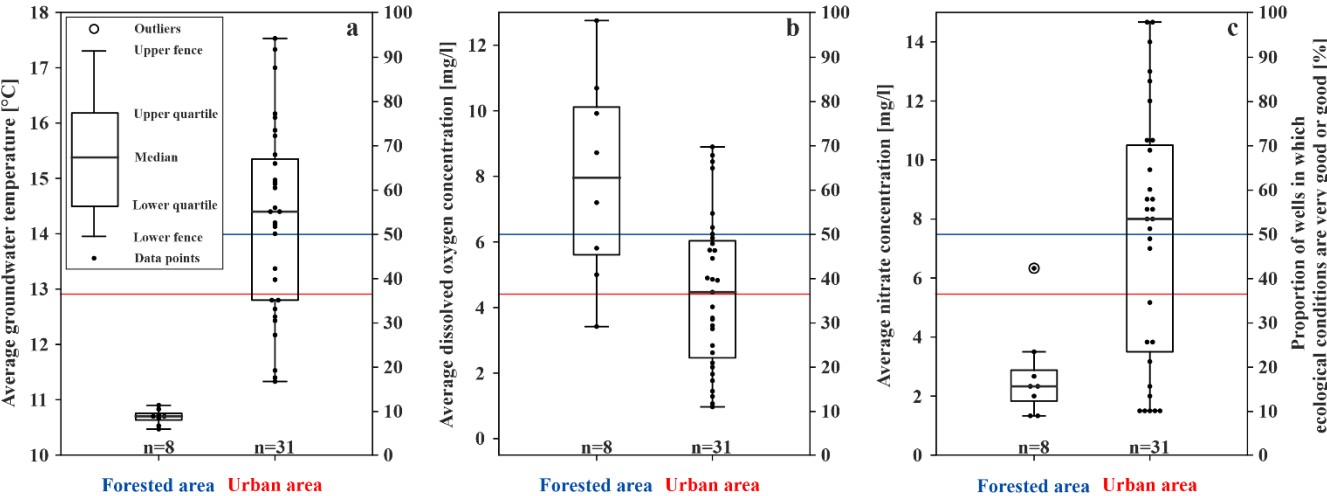

Figure 3: Boxplots of the physical and chemical parameters for the forested and urban area in the study site and the proportion of wells in which ecological conditions are very good or good in percentage [%] indicated by the blue (forested area) and red (urban area) lines (secondary axis); (a) average temperature of the repeated measurements [°C] at the bottom of the monitoring wells; (b) average content of dissolved oxygen [mg/l] of the monitoring wells; (c) average nitrate content [mg/l] of each monitoring well. (n = number of wells)

The content of dissolved oxygen acts as a limiting factor for groundwater fauna, since groundwater is usually under-saturated with a varying oxygen content between 0 and 8 mg/l (Griebler et al., 2014; Kunkel et al., 2004). In this study, the average content of dissolved oxygen in all wells is between 1.0 and 12.8 mg/l (Figure 3b and Figure S1a). As expected, the monitoring wells located in the forested area (Hardtwald) show the highest content, while the lowest values are found in urban areas and is likely linked to aquifer contamination and other anthropogenic effects (content of dissolved oxygen of forested vs. urban area: U-test: $p$-value = $5.3 \times 10^{-3}$, n = 8; 31). Urban water can be polluted in multiple ways, which affects the chemical and biological oxygen consumption in the groundwater. The higher the pollution and/or biological activity, the lower the dissolved oxygen (Kunkel et al., 2004; Griebler et al., 2014). Moreover, it seems that with a greater depth of the measurement wells the content of dissolved oxygen is increasing (U-test: $p$-value = $<10^{-13}$, n = 39). This can be explained by the fact that shallow wells can have a low water column in which oxygen can rapidly be consumed by groundwater microorganisms, chemical reactions and/or groundwater fauna. In the upper unscreened part of deeper wells, dissolved oxygen can be consumed while in the lower screened part oxygen is continuously refilled by oxic groundwater from the surroundings (Malard et al., 2002). Furthermore, reducing conditions in the overlaying soil can result in a low content of dissolved oxygen in groundwater.

Nitrate is often named as an important pollutant in groundwater. The natural and geogenic concentrations of nitrate in groundwater is usually under 10 mg/l (Griebler et al., 2014). In our study area, the average nitrate contents of all wells vary between 1.3 and 14.7 mg/l. In the urban area average nitrate concentrations are generally higher and correlate with the content

of dissolved oxygen (U-test: $p$-value = $4.0 \times 10^{-3}$, n = 39) showing the link between nitrate content and oxygen consumption. Wells with a content of dissolved oxygen below 1.5 mg/l have an average content of nitrate of 1.5 mg/l, most likely caused by nitrate reduction under anoxic conditions. Groundwater with reducing conditions (< 5 mg/l dissolved oxygen) has an average nitrate content of about 7 mg/l in contrast to groundwater with oxidising conditions with 9 mg/l, which promotes the oxidation of ammonium to nitrate. The lowest nitrate concentrations are found in the forested area (Figure 3c and Figure S1c), where atmospheric nitrogen is held back by forest soils (U-test: $p$-value = $1.7 \times 10^{-3}$, n = 8) and fertilization is prohibited due to water protection regulations in the forested area (Aber et al., 1998; Schönthaler and von Adrian-Werburg, 2008). Moreover, the average concentration of iron and phosphate are low and in most cases below the detection limit of the test (Figure S1d, e) and also below the natural and geogenic concentrations (phosphate: 0.05 mg/l (Griebler et al., 2014) and iron: 3.3 mg/l (Kunkel et al., 2004)) within the study site.

Considering these findings, clear differences in the spatial distribution patterns of abiotic groundwater characteristics are noticeable. The forested area shows lower average GWT than the urban area (U-test: $p$-value = $3.3 \times 10^{-5}$, n = 8; 31), lower nitrate concentrations (U-test: $p$-value = $4.1 \times 10^{-3}$, n = 8; 31) and higher dissolved oxygen concentrations (U-test: $p$-value = $5.3 \times 10^{-3}$, n = 8; 31), which indicates a correlation between abiotic groundwater characteristics and land use in the study area. Moreover, no impact of groundwater originating from the urban area on the wells in the forested area is observed, as the groundwater flow direction in Karlsruhe is northwest (see Chapter 2.1 and Figure 2c). Further investigations demonstrated that besides one larger and two smaller contamination sites (however, still with concentrations below the threshold values, Figure S1b), only minor groundwater pollution is documented in Karlsruhe (see Supplement). The chemical and physical parameters considered in the long-term monitoring system are within the range of local background and below threshold values of the drinking water ordinance of Germany (see Supplement for more information). Thus, the main documented impacts on groundwater quality in the study area are related to temperature and oxygen.

### 3.2  Groundwater fauna

The biotic communities of the groundwater consist of microorganisms and invertebrates (in particular Crustaceans) (Griebler et al., 2014). In the pool of samples, 3,666 individuals were detected in 37 of 39 wells, which means that 95 % of the wells are colonised (Table S3). With 2,047 individuals, the group of *Crustacea* was found to be the most abundant (56 %). 976 individuals (27 %) of the order of *Cyclopoida* dominated this group, followed by the genus *Parastenocaris* with 599 individuals (16 %), by the order of *Bathynellacea* (371), *Amphipoda* (66), *Harpacticoida* (33) and *Nauplia*. The communities of the monitoring wells also frequently contained Oligochaetes (1,343 individuals, 37 %). Furthermore, individuals of the phylum *Nematoda* (228 individuals) and *Microturbellaria* (46 individuals) were also often present.

Overall, there is a noticeable difference in the spatial distribution of species within the study area. Individuals of the subphylum *Crustacea* were found in larger numbers, with regard to the number of wells, in the monitoring wells in the forested area (690 individuals in eight wells) compared to those in the urban area (1,357 individuals in 31 wells). Furthermore, no individuals of the order *Bathynellacea* and only 135 individuals of the genus *Parastenocaris* were found in the forested area. In contrast,

larger numbers of the latter species as well as of Oligochaetes are characteristically found in the wells in the urban area.

However, in contrast to the abiotic characteristics, no clear pattern of faunal diversity and land use was observed as Crustaceans and individuals of other subordinate taxa were found both in the forested and in the urban area.

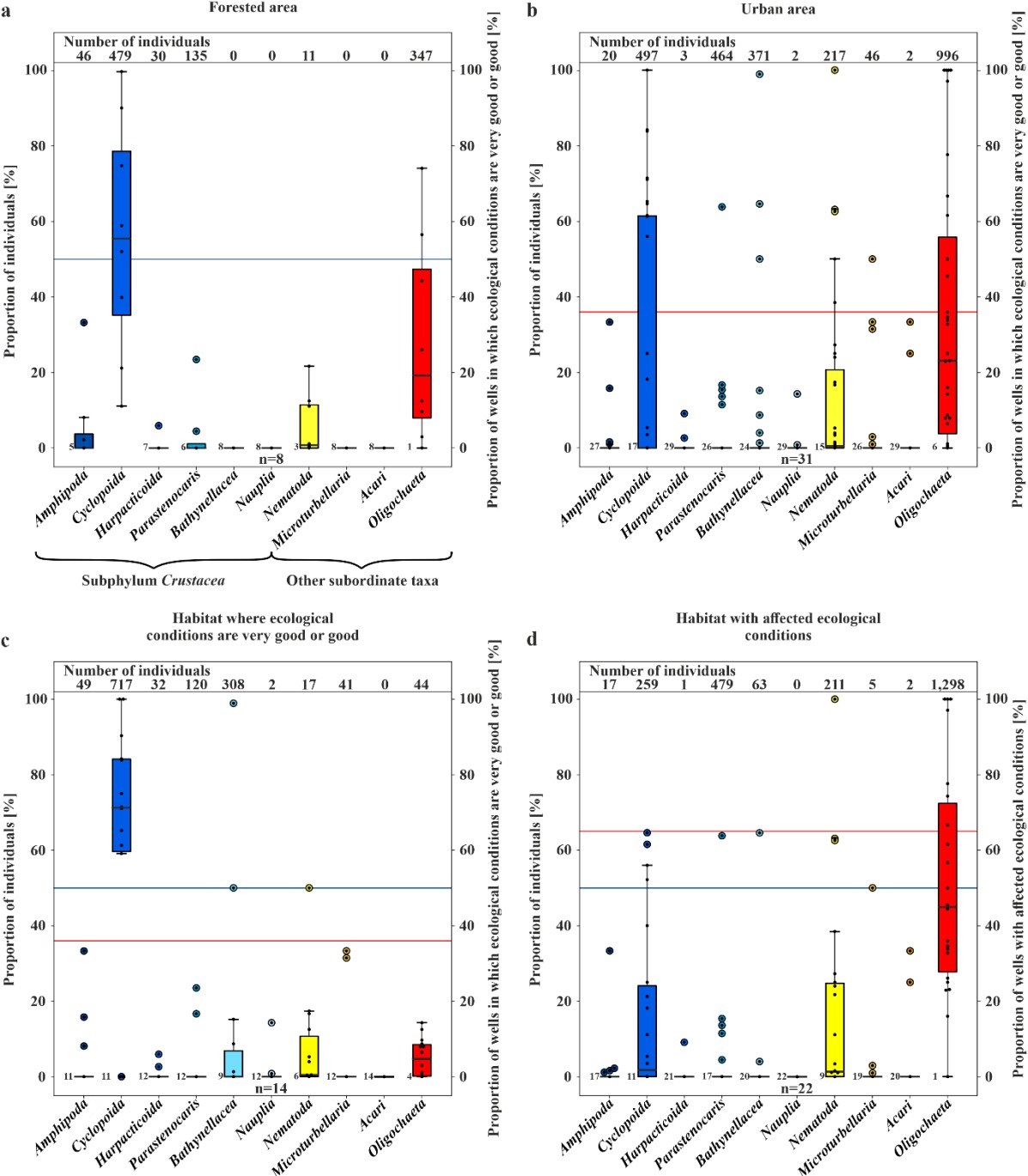

Figure 4: Boxplots of the amount of fauna [%]: (a) proportion of individuals and of wells in which ecological conditions are very good or good (secondary axis) [%] of the forested area; (b) proportion of individuals and of wells in which ecological conditions are very good or good [%] of the urban area; (c) proportion of individuals and of wells in which ecological conditions are very good or good [%] divided based on the results of the classification scheme by Griebler et al. (2014); (d) proportion of individuals and of wells with affected ecological conditions [%] divided based on the results of the classification scheme by Griebler et al. (2014). The colour of the boxes shows the different taxa in the samples. (n = number of wells)

Stygobiotic Amphipods, i.e. large-bodied invertebrates which due to their size have a habitat preference for open spaces such as wells (Table 1) (e.g. Hahn and Matzke, 2005; Korbel et al., 2017), were found in only three wells (Figure 2b). 46 individuals of this order were detected in the forest and 20 individuals in the urban area (Figure 4a,b). Although statistical analysis showed no clear differences between the abundance of Amphipods and land use (U-test: $p$-value = $1.5 \times 10^{-1}$, n = 8; 31), the higher number of individuals in the forest area could support the hypothesis that Amphipods indicate healthy groundwater ecosystems as they react most sensitively to disturbances such as pollutants (Korbel and Hose, 2011) and groundwater temperature. In laboratory experiments with a thermal tank, Brielmann et al. (2011) found that 77 % of the individuals of the studied Amphipods (*Niphargus inopinatus*) preferred areas with a temperature between 8 and 16 °C. In addition, Spengler (2017) and Issartel et al. (2005) observed maximum temperatures of up to 17 °C. The lack of a statistically significant correlation might also be related to the low number of wells (n = 8 in the forested area) and individuals (n = 46). Amphipods are important ecosystem service providers in terms of bioturbation and organic decomposition (Boulton et al., 2008). As observed in laboratory experiments (Smith et al., 2016), they actively move with migration speeds between 1.7 and $3.5 \times 10^4$ m per year. In most cases when Amphipods were found, higher concentrations of individuals of the order *Cyclopoida* were also identified (Abundance *Amphipoda* vs. *Cyclopoida*: U-test: $p$-value = $9.6 \times 10^{-5}$, n = 39). Individuals of the latter order were generally found in larger quantities in the majority of the wells (479 in the forested area and 497 in the urban area), as they are the largest group of Crustaceans in this environment (Fuchs et al., 2006) and can tolerate a wide temperature range (e.g. upper thermal limit of $26.9 \pm 0.2$ °C in laboratory tests by Sánchez et al. (2020))(Spengler, 2017).

The order *Harpacticoida*, which includes the genus *Parastenocaris*, have an elongated body shape and a stem-chiselling movement, which is why they are predestined for living in cavities and groundwater ( Hahn, 1996; Fuchs, 2007), preferring sand and gravel as a substrate (Galassi et al., 2009). Larger numbers of *Parastenocaris* (464 individuals), which can tolerate GWT from 8 to > 20 °C (Fuchs et al., 2006) (e.g. *Parastenocaris phyllura* up to 22.5 °C in laboratory tests (Glatzel, 1990)), were found in the urban area, especially in the northwest area (Figure 2b). This area is characterised by GWT between 16 and 18 °C, the highest at the study site. This observation is comparable with previous studies (Hahn, 2006; Hahn et al., 2013; Spengler, 2017), which showed that the genus *Parastenocaris* is particularly non-competitive and can often be found isolated in structurally burdened and physico-chemically altered areas. Accordingly, only 135 individuals were detected in the forested area.

In addition, quantities of *Bathynellacea* (371 individuals) were found in five monitoring wells all located in the urban area in a depth of 9.0 to 13.5 m at a GWT of 12-15 °C (Figure 4b). This order typically inhabits the interstitial groundwater, which is characterised by a dominant exchange with the surface water and high variations in GWT and can tolerate temperatures up to 18 °C (Stein et al., 2012). Interestingly, one location in the southern city area with 272 individuals is characterised by a high fluctuation in GWT (standard deviation of 3.4 °C) and a rather high nitrate content (8.3 mg/l) compared to wells in the forested area, which are both indications for a disturbed and stressed habitat.

Besides the group of Crustaceans, Oligochaetes, which can tolerate a wide temperature range, were also found in large abundance in the study site. A significant amount of the subclass *Oligochaeta* (996 individuals) was found in the urban area

(Figure 4b), compared to an overall number of 1,343 individuals. In general, the number of Oligochaetes is larger in locations with high GWT (12.6 – 17.3 °C) and nitrate concentrations up to 14 mg/l, which is above the geogenic concentration of 10 mg/l and higher compared to wells in the forested area.

Finally, Nematodes and Microturbellarians were found at locations with unfavourable living conditions, such as a low content of dissolved oxygen, or a high amount of fine substrates, as also reported by Hahn et al. (2013), both can tolerate high

temperature ranges (*Turbellaria*: 2 – 20°C (Herrmann, 1985), *Acari*: 9.1 – 18.5 °C (Więcek et al., 2013)). Here, both were found in larger quantities in the urban area of Karlsruhe (Figure 4b). This area has the lowest content of dissolved oxygen and relatively higher amount of detritus (> 2).

Eventually, correlation analysis between groundwater fauna and the chemical parameters showed that Stygobites are only slightly affected by groundwater chemistry (Hahn, 2006; Schmidt et al., 2007; Stein et al., 2010). Only the Spearman's rank

correlation coefficient $\rho$ between the number of taxa and the content of dissolved oxygen is significant with a value of $\rho = 0.55$ ($p$-value = $3.0\times10^{-4}$, n = 39). Moreover, it is assumed that groundwater fauna can usually cope well with short-term changes of chemical-physical parameters (Griebler et al., 2016). Previous studies showed that some species can even benefit from pollutants (Matzke, 2006; Zuurbier et al., 2013). In case of nitrate, numerous studies underline that nitrate at concentrations below 50 mg/l does not directly affect groundwater fauna (Fakher el Abiari et al., 1998; Mösslacher and Notenboom, 2000;

Di Lorenzo and Galassi, 2013; Di Lorenzo et al., 2020). As the highest average nitrate content per well is below 15 mg/l in this study, a direct negative effect of the nitrate concentration on the groundwater fauna is unlikely. Thus, nitrate is only mentioned as one measured parameter and is not discussed as a potential anthropogenic impact in this study.

The natural influence on porosity, groundwater flow and nutrient delivery were also discussed as a primary influence on natural Stygobites distribution in previous studies (Hahn, 2006; Korbel and Hose, 2015). One important natural influence is the local

geology, as fine sands and silts are typically rather harsh environments, resulting in an impoverishment of specific groundwater fauna such as *Crustacea* (Hahn, 1996). The city of Karlsruhe is located on carbonate ('Würm') gravel and river terrace sands, pervaded by bands of drifting sand and inland dune sands. These sediments are highly water-permeable and show almost exclusively vertical seepage of water movement. Flood sediments (on top of river gravel) and bog formations, are located in the east and west of Karlsruhe (Regierungspräsidium Freiburg, 2019). This local geology limits the cavity size and therefore

has impacts on the habitat of the groundwater fauna (Wirsing and Luz, 2007). For example, individuals of the genus *Parastenocaris* typically inhabit small-scale cavity systems (Spengler, 2017). Individuals of this genus can be found both in the wells drilled in gravel (4 wells) and in drifting sand sediments (3 wells) (abundance *Parastenocaris* vs. geological units: U-test: $p$-value = $1.4\times10^{-9}$, n = 39). Amphipods are predominantly found in measurement wells located in the 'Würm' gravels (in 5 of 7 wells) (abundance *Amphipoda* vs geological units: U-test: $p$-value = $9.0\times10^{-11}$, n = 39). Moreover, it seems that

differences in the geological units have an influence on the total amount of individuals (U-test: $p$-value=$1.7\times10^{-9}$, n = 39) and the relative amount of detritus (U-test: $p$-value = $3.0\times10^{-3}$, n = 39). As these results show, regional geology seems to have an influence on the occurrence of specific groundwater taxa and on the number of individuals as well as on food supply, in terms of available organic matter. However, it is not possible to give a reliable estimate of the strength of the anthropogenic impacts,

e.g. if they are strong enough to overrule the regional selective forces. Hence, this should be investigated in more detail in future studies.

Limitations regarding the sampling method must be considered when interpreting the faunistic results. In this study, a simple basic screening of well water was conducted using a net sampler and bailer to examine conditions in the groundwater monitoring wells (39 wells with an average diameter of 132.5 mm, which corresponds to an area of 0.003 ‰ of the total urban area). According to the sampling manual of the PASCALIS Project 'the use of a phreatobiological net alone is considered as a satisfactory method for sampling groundwater fauna in large diameter wells' (Malard et al., 2002). Yet, several studies (e.g. Scheytt, 2014) report that scooped samples of wells are not representative, and therefore the water remaining in a well has to be purged and discarded before sampling. Nevertheless, pumping can result in the selection of the taxa, especially in the presence of very fine sediments, and can result in changes of the sediment composition in the surrounding of wells and therefore in changes of habitat conditions. Other studies, on the other hand, found no significant differences in hydro-chemical values (temperature, pH, dissolved oxygen, etc.) between the surrounding groundwater and the standing water in a well (Hahn and Matzke, 2005; Korbel et al., 2017). The sampled groundwater fauna of corresponding wells and aquifers were also shown to be similar with respect to the types of faunal communities. However, in terms of total abundance, as well as the numbers of individuals per litre, monitoring wells appear to exhibit larger numbers caused by filtration effects (Hahn and Matzke, 2005; Hahn and Gutjahr, 2014; Korbel et al., 2017). As the aim of this study is to provide an overview of the groundwater fauna community (assess biodiversity) and to receive a first impression of groundwater ecology, sampling the fauna by using a net sampler is sufficient. In order to achieve a representative sampling of groundwater fauna in the aquifer and to reflect the occurring species at a community level, a more comprehensive sampling method is required, e.g. the use of a defined standard sampling method using a pump to collect animals (Malard et al., 2002). Care should also be taken when interpreting faunistic results of sites that are sampled in different years. To improve comparison of the biotic communities, a consistent sampling period of every well is necessary in the future.

### 3.3 Classification scheme by Griebler et al. (2014)

In three wells, evaluation with the classification scheme by Griebler et al. (2014) was not possible due to ocherous conditions in two monitoring wells and low content of dissolved oxygen (<1 mg/l) in the third well. According to the classification scheme by Griebler et al. (2014), unstressed (meaning no natural or anthropogenic stressors), or natural groundwater habitats have an amount of more than 70 % of Crustaceans and less than 20 % of Oligochaetes. In 36 % of the sampled wells, i.e. 14 out of 39, these criteria were fulfilled indicating very good or good ecological conditions or in other words a natural groundwater habitat (Figure 4c). These natural areas tend to contain more individuals of the orders *Amphipoda, Cyclopoida* and *Bathynellacea*. Monitoring wells, which do not fulfil these criteria and are accordingly defined as affected areas not having natural ecological conditions, contain more Oligochaetes and also Nematodes, which is partly explained by the used criteria of this classification scheme (Figure 4d).

Surprisingly, only 50 % of the wells in the forest, which is also the catchment area of the drinking water supply of Karlsruhe, are described as natural groundwater habitats. An identical number of wells yielded habitats with affected ecological conditions. The main difference between natural and affected wells in the forested area arises from the occurrence of specific species. 86 to 100 % of species found in natural wells are Crustaceans, in contrast to affected wells with only 33-67 % (Table S2 and Table S3). However, the abiotic parameters scarcely differ between natural and affected wells (average values for GWT: 10.8 and 10.6 °C, dissolved oxygen: 7.1 and 8.8 mg/l, nitrate: 2.5 and 3.0 mg/l), indicating that there are other processes or parameters that influence the groundwater fauna in these wells. One reason could be the varying local geology as mentioned above. Moreover, food supply is one of the most limiting parameters for the survival of groundwater fauna (Datry et al., 2005; Hahn, 2006). If the organic carbon supply varies on a small scale, this can influence microbiology and therefore groundwater fauna as well, although, short-term changes in nutrient supply can be compensated by groundwater fauna.

In contrast to the forest land, the majority of wells (65 %) in the urban area are categorised as affected habitats. As expected, this indicates anthropogenically influenced groundwater ecosystems beneath the studied urban area. Once more, no significant differences between the abiotic parameters of natural and affected wells are observed (e.g. median of dissolved oxygen: 4.7 and 5.8 mg/l, median of nitrate: 7.2 and 7.8 mg/l). On the other hand, the remaining 35 % of the wells in the urban area show natural ecological conditions even though some of them are located in areas with anthropogenic impacts such as increased groundwater temperatures. Hence, no distinct spatial pattern of the ecological condition with respect to land use could be identified.

In future, a further subdivision of a study area in more land use categories could be beneficial to specifically look at typical anthropogenic impacts. Furthermore, the integration of more biological criteria is useful to improve the results of the assessment according to Griebler et al. (2014). Because of heterogeneous groundwater ecosystems in Germany it is likely that reference values provided by Griebler et al. (2014) do not reflect the situation in Karlsruhe correctly. Considering site-specific characteristics and reference values would lead to a more robust assessment. Other assessments, like the similarly structured GHI or wGHI[N] (Korbel and Hose, 2017; Di Lorenzo et al., 2020b) can additionally be used. Moreover, there are a couple of newly developed indexes, like the D-A-C-Index, which is based on microbiological indicators and shows whether groundwater reserves deviate from natural references (Fillinger et al., 2019), which can be used in the future. As mentioned in the introduction, another way to quantify the relevant ecological conditions in the groundwater is the GFI. During the preparation of this study, the GFI was tested on the data (see Supplement), however, it did not provide any additional information or valuable insights and was therefore excluded. The influence of multiple stressors, such as the pollution of the groundwater through industrial plants etc., and their effects on the governing parameters can bias the GFI. In general, the GFI seems to be suitable only for unpolluted and anthropogenically undisturbed groundwater with sufficient oxygen concentrations (> 1 mg/l). Moreover, under urban areas changes in GWT are caused by anthropogenic heat inputs (Menberg et al., 2013b, 2013a; Benz et al., 2014; Tissen et al., 2018), rather than being related to surface water influences. Hence, the GFI appears to be unsuitable for the assessment of the groundwater fauna in an urban setting. The same outcome emerges for the Shannon diversity index,

which was also tested during the preparation of the study and showed no clear distribution pattern according to faunal diversity and was therefore not considered further.

### 3.4 PHATE analysis

A PHATE analysis is conducted using the following 15 input parameters: depth, GWT, nitrate and phosphate content, relative amount of detritus, geological unit, numbers of taxa, number of individuals, Shannon diversity, amount of Crustaceans and Oligochaetes (according to Griebler et al., 2014) and the abundance of Amphipods as well as of individuals of the order *Cyclopoida*, *Bathynellacea* and the genus *Parastenocaris*. The content of dissolved oxygen is not considered in this analysis, since it was always above the limit of 1 mg/l, except for in one case. Thus, dissolved oxygen is not expected to have an influence on the groundwater fauna in our study area.

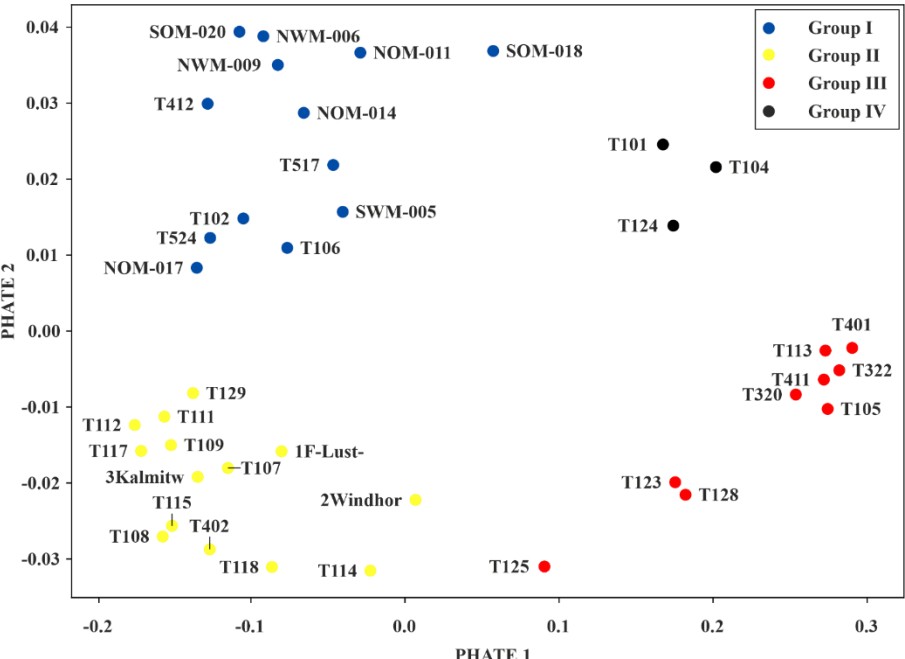

**Figure 5: PHATE visualization showing similarities between measurement wells. Different colours indicate the four clearly separable groups.**

Four groups, which can be assigned predominant characteristics, can be distinguished in the PHATE visualization (Figure 5, Figures S3-S4). Three measurement wells (Group IV) contain neither Oligochaetes nor Crustaceans, indicating unfavourable living conditions. In contrast, the nine wells of Group III contain high amounts of Oligochaetes (100 % Oligochaetes according to the scheme of Griebler et al. (2014)), and an average GWT of 14.3 °C (Table S4). However, diversity and abundance was found to be low in Group III.

An even higher average GWT of 15.0 °C was found for Group II, which mostly consists of wells drilled in drifting sand sediments. Surprisingly, these wells also show the highest diversity (≥ three Taxa per well), the highest Shannon diversity (see

Supplement), highest amount of individuals in total, as well as of individuals of the genus *Parastenocaris*. Individuals of this genus are often found isolated in altered areas (Spengler, 2017). Moreover, in five wells of Group II individuals of the order
*Bathynellacea*, which can tolerate temperatures up to 18 °C and typically inhabit interstitial groundwater (Stein et al., 2012), were found. The presence of individuals of the genus *Parastenocaris* and the order *Bathynellacea* in Group II suggests that they may act as type species for urban situations. The observation that Group II shows the highest GWT and the highest Shannon diversity is in contrast to findings of previous studies that noticed decreased diversity at elevated temperatures (Brielmann et al., 2009). These diverging observations suggest that faunal quantities, such as diversity or abundance, are not
always suitable indicators for changes within organism communities. For example, if species disappear due to increased temperatures and are substituted by more tolerant species, the difference in diversity may be marginal and the change in the community may not be noticeable.

Wells of Group I (blue) are drilled predominantly in Würm gravel (geological unit of Group I vs. Group II: U-test: *p*-value = $8.2 \times 10^{-3}$, n = 13; 14), while having the lowest GWT (GWT of Group I vs. Group II: U-test: *p*-value = $2.0 \times 10^{-5}$,
n = 13; 14). These wells show a moderate diversity and amount of individuals, yet the highest average amount of Crustaceans as well as the highest amount of Amphipods and individuals of the order *Cyclopoida*. Considering these findings and the U-Test results (see Table S5), the grouping of the measurement wells seems to be influenced by the composition of the groundwater organism communities, the faunal diversity (numbers of taxa and amount of individuals), as well as the geological unit and the GWT (Figure S3-S4).

Considering the spatial distribution of the grouped wells in the study area, it becomes apparent that all wells in the forested area fall within Group I (Figure 5). Those wells which are located outside the forested area are in locations with nearby green areas (parks, recreational areas, etc.). In contrast, the wells of the other three groups are heterogeneously distributed within the urban area. Many of the measurement wells of Group III and IV are associated with suspected or known contaminated sites (Figure S1b). Overall, a spatial pattern of abiotic groundwater characteristics (GWT, nitrate content) and occurrence of
particular species (*Parastenocaris*) within the study area is apparent in the PHATE analysis, which confirm the classification according to land use. Yet again, no clear spatial pattern regarding faunal diversity in the study area could be identified. Although, a tendency of clustering of wells from Group III with higher diversity and amount of individuals can be seen in the northwest city area.

## 4. Conclusion

The aim of this study is to provide a first assessment of the ecological state of groundwater in an urban area and to distinguish areas with a natural state of groundwater ecology from anthropogenically affected areas. To achieve this, we examine the groundwater fauna, as well as abiotic parameters in 39 groundwater monitoring wells in residential, commercial and industrial areas (31 wells) and a forested area (eight wells) outside the built-up area of Karlsruhe, Germany, using the simple classification scheme by Griebler et al. (2014) to characterise the sampled monitoring wells.

We found a noticeable difference in the spatial distribution of abiotic groundwater characteristics and special species within the study area. The forested area shows lower GWT, lower nitrate concentrations and higher dissolved oxygen concentrations, which indicates a correlation between abiotic groundwater characteristics and land use. Moreover, Amphipods are more abundant in wells in the forested than in urban area. However, both in the rural forested and in the urban area Crustaceans and individuals of other subordinate taxa were widely found and therefore no clear spatial pattern regarding faunal diversity and

land use was found. In terms of faunal quantity, Crustaceans were found in larger numbers, with respect to the number of wells, in the monitoring wells in the forested area compared to those in the urban area. Larger amounts of the genus *Parastenocaris* as well as of Nematodes and Oligochaetes were found to be characteristics for wells in the urban area.

Furthermore, no clear spatial pattern of ecological groundwater conditions according to the classification scheme by Griebler et al. (2014) could be observed. Surprisingly, only 50 % of the sampled wells in the forested area were described as natural

(undisturbed) groundwater habitats, while the other four were characterised as habitats with affected ecological conditions. Yet, the majority of wells (65 %) in the urban area were classified as affected locations, suggesting that there are noticeable differences in the groundwater ecosystems between the surrounding forested and urban areas. The Level 2 assessment from Griebler et al. (2014) can help to achieve a more reliable and quantitative ecological assessment of urban aquifers as it divides groundwater ecosystems in ecological grades according to the intensity of anthropogenic disturbance. It is based on the use of

local reference values and the collaboration with experts, however, is challenging to apply. Therefore, further studies with large-scale and repeated measurement campaigns are needed to verify our findings. This should also include other cities and the determination of undisturbed local reference values which are required for a more reliable but also quantitative ecological assessment of urban aquifers. Moreover, a wider range of indicators should be considered in a classification scheme, such as temperature, porosity of the aquifer, groundwater flow, pollutants and nutrient supply, especially when investigating urban

areas. In addition, an important adaptation for an improved evaluation method is the determination of fauna at species level which will provide more information (i.e. about Stygobionts, Stygophiles, Stygoxenes) and also consider the endemism of stygobiotic species. In this context, classification schemes should pay more attention to the different groundwater species and their potential use as indicator species.

Finally, city and energy planning should seriously consider urban groundwater ecosystems as they provide valuable

information for a sustainable use of the subsurface.

**Data availability**

**Team list**

Institute of Applied Geosciences (AGW), Karlsruhe Institute of Technology
Prof. Dr. Philipp Blum (philipp.blum@kit.edu)

MSc. Fabien Koch (fabien.koch@kit.edu)

Dr. Kathrin Menberg (menberg@kit.edu)

MSc. Svenja Schweikert (svenja.schweikert@googlemail.com)

Faculty of Nature and Environmental Sciences, University Koblenz-Landau

Dr. Hans Jürgen Hahn (hjhahn@uni-landau.de)

Dr. Cornelia Spengler (spengler@uni-landau.de)

**Author contributions**

PB and HJH provided the topic and supervised the work, together with KM. SS and CS executed the field work and evaluated the samples. FK evaluated the collected data and interpreted as well as visualised the results and wrote the first draft of the

paper. KM, CS, HJH and PB participated in editing the paper.

**Competing interests**

The authors declare that they have no conflict of interest.

**Acknowledgements**

We would like to thank Annette März (Environmental Service, City of Karlsruhe), Michael Schönthal (Public Utilities

Karlsruhe) and Friedhelm Fischer (Civil Engineering Office of Karlsruhe). Special thanks are also given to Christine Buschhaus and Tanja Liesch for their support with the measurement and sampling (Institute of Applied Geosciences, Karlsruhe Institute of Technology).

We acknowledge support by the KIT-Publication Fund of the Karlsruhe Institute of Technology.

**Financial support**

This research did not receive any specific grant from funding agencies in the public, commercial, or not-for-profit sectors.

**Review statement**

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
