# Peer review of "Groundwater fauna in an urban area: natural or affected?"

_Hydrology and Earth System Sciences, 2020_

## Referee Comment (RC1) · Anonymous Referee #1 · 8 Jun 2020

This article is an application of an existing method to assess groundwater ecological condition. The article utilises a classification scheme based on a single threshold of proportion of crustaceans and oligochaetes within sample wells, with varying success. The manuscript acknowledges several limitations of using this single method suggesting that multiple methods should be used to fully understand impacts of humans on groundwater ecology. The research presented increase awareness of groundwater ecosystems and the threats facing them, however requires further analysis to justify some of the claims made. As such, I recommend major revision, purely because of the requirement for further statistical analysis.

General Comments:

Generally, the sections flow well and it is easy to understand. The manuscript needs

to be thoroughly edited as there are multiple issues with grammar, and the manuscript can be reduced in length particularly in the introduction. The figures and table are well presented. The methods and results section needs to have some aspects clarified. There is a lack of statistical analysis throughout the manuscript which detracts from the quality of the paper. The results show some interesting trends in the distribution of biota, however without the necessary statistical analysis of this data, it is difficult to establish if there are significant differences between landuses, or if these trends are just due to differences in sample size (n8-n31) between the two landuses. This needs to be addressed, as currently there are speculations that differences in means indicates differences between landuses without any specific statistical analysis. A simple ANOVA or t-test would, in most cases, suffice and allow a more thorough analysis of this useful data.

Specific comments:

Introduction In general the introduction is a little too long and can be made more concise. Eg paragraph starting line 45 and line 50 could be compressed and merged. Line 35-37: Whilst these may be the usual temperatures for stygofauna within the region of this study, they exist in temperatures well over 14-16 deg on a global basis. This sentence needs to be rephrased. Line 38: Remove 'the' from "the German and European legislation" Line 44 remove 'data recorded by' in the brackets Line 54: Typo error (Protocol for the Assessment. . ... Line 83: Korbel & Hose 2011 is correct reference, also consider Di Lorenzo et al. 2020 Ecological Indicators, 116, 106525.

Methods

Line 116: replace 'with' to 'was' Line 116-117: improve sentence structure 'is mainly caused by' is incorrect, consider 'which' is mainly caused by. . . or rewrite sentence appropriately. Line 120-124: condense and combine sentences Line 139-140: belongs in the results section not methods Section 2.3. I found this section hard to read, particularly due to some grammatical errors. This section is too long and verbose, it needs to

be rewritten to make it clearer. The second paragraph starts well. I suggest removing the sentence start on line 160 "this requires to obtain also…."

Statistical analysis: You do not mention any of the statistical analysis competed in this paper. To be able to distinguish between forest and urban areas, you should at a minimum be completing some statistical analysis of the water quality data you have collected, even if this is simple ANOVA or t-test analysis. This is a major issue that detracts from the quality of this paper. I understand that you have used average values of the sampling wells, however determine whether there are statistical differences between (for instance) temperature at forested areas in comparison with urban areas, and look at the relationships between temperature, well depth and landuses. This analysis would greatly improve the scientific credibility of this study.

Results/Discussion

Line 180: complete statistical analysis to indicate if there are significant differences in temperature between urban and forests areas- it appears that there are. Lines 192: while the box plots show that there are differences between forests an urban areas in DO and nitrate with landuses, these do not appear to be statistically different. I am not convinced that there are differences in DO and Nitrate between landuses this needs further discussion, as does the large differences in n values between the landuses. Line 196: References in chronological order Line 201: 'hold back' should be 'retained' or 'held back' Line 201: suggest these sentences are combined and reduced eg '…..where atmospheric nitrogen in retained by forest soils and fertilization is prohibited due to water protection regulations' Line 207-209: Again you cannot claim 'clear differences' without adequate statistical analysis of these factors. You need to run further analysis of the data for this statement. Line 231: I would not say that amphipods 'predominantly live within wells' rather they have a habitat preference for open spaces such as wells. Paragraph starting line 231: It appears that amphipods are significantly higher in forested areas than urban areas, however without analysis this cannot be determined. This may be ecologically important and should be discussed. It is also

worthwhile looking at the correlation between cyclopoida and amphipods as briefly mentioned in line 238. Line 238-240: Incorrect grammar... remove 'be' Line 248: Incorrect grammar Line 274- 280: The issue of purging wells needs further discussion as this is a limitation of your study. If you are looking at proportions of crustaceans to oligochaetes this is almost certainly affected by sampling method. The sentence on line 277 needs to indicate that relative abundances and proportions of crustaceans is likely to be impacted by the sampling methods, thus caution must be taken when interpreting the results. Lines 295 -300: Could this also be due to organic carbon supply? Would level 2 assessment clarify these issues if it were undertaken? Line 305: Could the high (35%) of urban areas displaying natural sites be due to the sample methodology; ie were they classified as good incorrectly due to high proportions of crustaceans that may be influenced by the lack of purging of the wells?

Conclusion:

The conclusions of this study to me indicate that the method you have adopted (ie net sample wells and use the proportions of oligo/crustacean populations to determine ecosystem condition) need to be investigated further. The disproportionate number of crustaceans in wells due to sampling methods may be impacting the assigning of "OK" condition to sites that are actually impacted. Potentially a wider range of indicators need to be used including expanding on the use of only oxygen concentration in the classification scheme. The Level 2 assessment (Figure 1) also needs to be discussed in the conclusion.

---

## Referee Comment (RC2) · Anonymous Referee #2 · 9 Jun 2020

**General comments**

This study on the distribution of groundwater fauna in the shallow subsurface of urban (city of Karlsruhe) and rural (nearby forest) areas, as well as the use of groundwater fauna for the assessment of the ecological status in groundwater has considerable scientific novelty. To my knowledge, this is the first study that investigates groundwater ecological aspects in a city's subsurface. This strength by novelty, however, is kind of counteracted by serious weaknesses. While I like the study very much on one hand, it is a pity that the authors did not spend enough time to distill the best out of it. Besides obvious shortcomings in the study design (the selection of chemical parameters measured, the restriction to only well water), the authors did not dig at all into the data available in a 'statistical' sense. I do see more categories of land-use types. I

think measures such as well depth and origin of groundwater (what if groundwater impacted in the urban area travels underneath the forest where it is sampled) would be interesting aspects to evaluate. Moreover, fauna data set comes along with further information that has not been used, i.e. the Shannon-Wiener biodiversity and the ratio of stygobites/stygophiles vs. stygoxenes. Not to talk about the determination of individual 'species' of Crustaceans and other groups of animals that could resolve the picture much more. Although, the basic water chemistry in the urban groundwater exhibits some differences to the groundwater sampled in the rural area, there is obviously no clear indication for a 'contamination' of the urban groundwater. An exception is only some temperature deviations. Thus, why it is expected that the groundwater fauna in the urban area is different. I would have loved to see a few hypotheses that are tested. When reading the preprint I also got the impression that most groups of groundwater fauna are described as quite temperature tolerant, however, other publications of the same authors claim the strong sensitivity of groundwater fauna upon groundwater warming. I really missed individual statistical testing of such questions. To be very honest, the paper addresses a really interesting topic that ghas ahrdly been studied to date but has not been properly prepared before submission. My feeling is also that some of the co-authors have not spend much time with the paper, otherwise it would not contain so many flaws. In the following, I will try to provide detailed comments that may help to improve the manuscript. Overall, I not sure if the paper, even when reworked properly, will satisfies the high standard of HESS.

**Specific comments**

P1 L19-20: How have the anthropogenic impacts be measured. I agree that elevated temperature may be seen as Impact. What else? The groundwater chemical analyses do not focus on any contaminants, with exception of nitrate; and nitrate concentrations are not elevated.

P1 L21: it is mentioned here that more comprehensive assessment methods are required to fully capture the different effects on groundwater fauna. I agree. However,
you should mention, at least in the discussion section, what you think of.

P2 L44-50: This paragraph does not seem to be linked to the what is introduced before and after.

P2 L51: There is a pile of studies dealing exactly with that. You should name some as examples. What is really new with your study is that there is hardly anything investigated in urban areas.

P2 L53: Are you sure that the European Union (FP5) PASCALIS project focused on 7 North-American regions. Please check that again.

P3 L57: If you state here that regional features have a stronger influence on groundwater fauna than local habitat features, you should test that with your data set. If this is true, maybe the anthropogenic impacts are not strong enough to overrule the regional selective forces. This point should also e discussed.

P3 L83: you could also have used a different approach to look at your results. What if you treat the forest samples as your local natural reference? Just an idea. Starting from there, you could evaluate which well downtown Karlsruhe match natural conditions and which not. Currently, you obviously use a German-wide reference conditions and thresholds (Crustaceans >50%, worms

the recommendations of Hahn & Gutjahr published in 2014 when sampling took place between 2011 and 2014.

P5 L134: what you mean with 'integrative sampling'? Explain!

P5 L136: replace 'groundwater ecology' by 'groundwater ecological status' ; we 'sampled' the fauna ...

P6 L145: If this table shall stay in the paper then the information provided with the individual groups of organisms asks for a balancing. The provided information is very heterogeneous. Some of the terms used have not been explained before, e.g. 'sty-gophile'.

P7 L156: No, I do not agree at all. There is many natural groundwaters in good ecological shape that do not contain any dissolved oxygen, they may also produce ochre where they come in contact with oxygen. I guess we agree that these sites are not 'good' habitats for groundwater fauna. However, the absence of fauna does not necessarily mean a disturbed ecosystem status.

P7 L170: Doesn't it make sense to further categorize the land use types, also within the city limits?

P9 L177: did you also consider well depth in your data analysis. It si a big difference between 8.5 and 39m below land surface which may affect occurrence of fauna and the availability of dissolved oxygen.

P10 Figure3: why are there two lines (red and blue) indicading the percentage of wells with good and affected ecological status? Automaticly one looks if the box is above or below. However, the values of the individual physicaö-chemical parameters are not in line with the ecological status. I recommend to delete the lines.

P10 L192: To my understanding, a concentration of 1 mg/L dissolved oxygen in wells water strongly indicates that there are anoxic conditions in groundwater. As is mentioned I the preprint well water is not representative for groundwater. To my opinion,

HESSD
since well water is open to the atmosphere, DO concentrations are likely to be overestimated. Gw fauna may, at times of elevated DO in groundwater migrate through the local subsurface and enter wells. There, they may outlast times of no oxygen in the surrounding aquifer. Frankly speaking, I am not sure if the threshold of 1mg/L of DO mentioned before should refer to the surrounding groundwater.

P10 L198: The study does not show high nitrate concentration! When stating that  $\leq$ 10 mg/L is natural, then in consequence nitrate concentrations between 1.3 and 14mg/l are not high!

P10 L199: In general, their relationship between DO and nitrate is not inversely correlated. Only when the oxygen is gone nitrate is reduced. As such, low or no oxygen goes along with low or no nitrate. I do not get the 'link' between oxygen and pollution claimed here (P10 L200).

P11 L220: does this mean that Parastonocaris and Bathynellacea are 'type'-species (groups) for urban situations? Such a possibility is not discussed in the paper. There are groundwater ecology experts in the list of authors. I miss an in depth interpretation of the ecological data.

P13 L231: Is it that stygobiont amphipods live predominantly within wells? This is to my opinion not a correct interpretation of what is published in Hahn & Matzke (Hahn is co-author of this preprint) and Korbel et al.

P13 L264: When the authors write about 'groundwater quality' it is not straightforward what is meant. Only very basic water chemistry (e.g. selected nutrients, pH, DO) and temp was measured. There is no indication for a 'bad' or 'impacted' groundwater quality (with the exception in temperature), so why should the groundwater fauna show associated distribution patterns.

P13 L235: Does the study of Brielmann et al. 2011 really state that amphipods react sensitive to a gw temperature of 11  $\pm$  5°C (which is natural gw temp in central Europe)

HESSD
or do they refer to a change of ambient gw temp by 11°C? Check that carefully.

P14 L266: It would help the reader if you indicate the general groundwater direction in one of your maps (Fig. 2). If the groundwater flow direction in the area is north-west, then it is very likely that groundwater originating from the urban area is travelling below the forest. This point should be discussed as well.

P14 L285: Why only the two criteria (>70% Crustaceans and

Agency (UBA) but the result from a UBA funded research project. This makes an important difference. The funding agency not necessarily identifies itself with the outcome of funded projects. The 'invention' and 'responsibility' is with the authors from the study. As such, I would not call the scheme used, and UBA classification scheme. Same applies to P2 L31.

P1 L16: wrong wording: 'fine' ecological conditions. Replace by 'good', 'natural' or something similar. Best you use the terminology used with the assessment scheme you used.

P2 L26: HESS is an international journal. I would cite 'German' and 'grey' literature only if there is not similar publication in international journals. This is my very personal opinion.

P2 L28: 'retention' is the wrong term here! What you mean is 'degradation' or 'mineralization'.

P2 L30: delete 'valuable'. What do you mean with 'tied'? Reword.

P2 L34: change 'relatively' to 'typically' or 'naturally'. Typo: Brielmann et al. 2011 not 20011.

P2 L35: change Žstygobiote' to Žstygobite' or 'stygobiont'.

P2 L38: ... groundwater ... is not yet recognized as a protected habitat ... reword this part of the sentence. Wat you probably mean is that gw is not yet recognized as an ecosystem that deserves protection.

P2 L39: change 'assessing groundwater ecology' into ' assessing groundwater ecological status'.

P3 L59: delete 'Unfortunately'. Not needed.

P3 L65: do the temp fluctuations range between  $4^{\circ}C$  and  $20^{\circ}C$  or is there a temp fluctuation with a temp range between  $4^{\circ}C$  and  $20^{\circ}C$ . Try to be more precise with your

**HESSD**
wording.

P3 L73: change 'clearly increasing' to 'increased and 'usually decreases' to 'decreased'.

P3 L75: Brielmann et al. 2011 not 20011!

P3 L79: The UBA did not develop anything! The UBA funded a research project in which these tools you refer to were developed. Rephrase this sentence.

P4 L96: change 'waterside filtration' to 'river bank filtration'.

P4 L98: 'beneath' an urban area

P4 L99: you can not sample thermal properties. You collected or sampled gw fauna and 'analyzed' gw chemistry and measured gw temp.

P4 L100: Again, it is not the classification scheme of the UBA.

P4 L101: 'state of ecosystem quality' sounds weird.

P4 L116-117: annual mean LST! Is this what you mean?

P4 L120: Didn't you specifically 'analyze' statistically if wells in the area of known contaminations show different features than others?

P7 L149 ... classification scheme in the framework of a research project funded by the ...

P7 L152: O.K. is an improper term in this connection

P11 L207: chemical characteristics do not distribute! There is distribution patterns.

P13 L242: 'Larger'!

P13 L252: 8.3 mg/l is 'not' a rather high nitrate content! Same applies to P13 L257.

151, 2020.

---

## Author Comment (AC1) · 30 Jul 2020

Dear referee,

we would like to thank you for your time and the constructive comments, which helped to improve the quality of the manuscript. Please find our detailed replies on the comments below. We hope that we answer all your remarks.

Referee #1:

This article is an application of an existing method to assess groundwater ecological condition. The article utilises a classification scheme based on a single threshold of proportion of crustaceans and oligochaetes within sample wells, with varying success. The manuscript acknowledges several limitations of using this single method suggest-

ing that multiple methods should be used to fully understand impacts of humans on groundwater ecology. The research presented increase awareness of groundwater ecosystems and the threats facing them, however requires further analysis to justify some of the claims made. As such, I recommend major revision, purely because of the requirement for further statistical analysis.

Response: We partially agree. Thus, we performed a more profound statistical analysis (e.g. U-tests), which are presented below in our replies to the 'specific comments'.

General comments

Generally, the sections flow well and it is easy to understand. The manuscript needs to be thoroughly edited as there are multiple issues with grammar, and the manuscript can be reduced in length particularly in the introduction. The figures and table are well presented. The methods and results section needs to have some aspects clarified. There is a lack of statistical analysis throughout the manuscript which detracts from the quality of the paper. The results show some interesting trends in the distribution of biota, however without the necessary statistical analysis of this data, it is difficult to establish if there are significant differences between landuses, or if these trends are just due to differences in sample size (n8-n31) between the two landuses. This needs to be addressed, as currently there are speculations that differences in means indicates differences between landuses without any specific statistical analysis. A simple ANOVA or t-test would, in most cases, suffice and allow a more thorough analysis of this useful data.

Response: We agree on the grammar issues. Hence, the manuscript was carefully revised by a native speaker to ensure correct English.

We agree on the length of the introduction. Several sentences were shortened or even deleted (for more details see the marked manuscript attached).

We partially agree on the statistical analysis as already mentioned earlier, which is

detailed below (Comment #12).

Specific comments

Comment #1: Introduction: In general the introduction is a little too long and can be made more concise. Eg paragraph starting line 45 and line 50 could be compressed and merged.

Response: We agree on the length of the introduction and the conciseness. Thus, we shortened or deleted sentences, e.g. we condensed the paragraph starting in line 47: "A closer look at the German federal state Baden-Württemberg is given in the study by Hahn and Fuchs (2009), which focuses on defining stygoregions based on different hydrogeological units. They conclude that the observed patterns of groundwater communities reflect a high spatial and temporal heterogeneity of aquifer types with respect to habitat structure, food, oxygen supply etc."

Comment #2: Line 35-37: Whilst these may be the usual temperatures for stygofauna within the region of this study, they exist in temperatures well over 14-16 deg on a global basis. This sentence needs to be rephrased.

Response: We agree on the reformulation of this sentence (lines 35-37). Nevertheless, it was not possible to find international values. Thus, we clarified the spatial reference of the values in this sentence: "Hence, they are assumed to be cold stenotherm, which means that they prefer cold temperatures and can hardly persist water temperatures over 16 °C (Brielmann et al., 2009) or rather 14 °C (Spengler, 2017) in Central Europe for a longer time period."

Comment #3: Line 38: Remove 'the' from "the German and European legislation"

Response: We agree. Done.

Comment #4: Line 44 remove 'data recorded by' in the brackets

Response: We agree. Done.

Comment #5: Line 54: Typo error (Protocol for the Assessment: : :..

Response: We agree. Done.

Comment #6: Line 83: Korbel & Hose 2011 is correct reference, also consider Di Lorenzo et al. 2020 Ecological Indicators, 116, 106525.

Response: We agree and now consider this new study in the manuscript. Hence, we added the following paragraph (see lines 91-93 in the marked manuscript):

"This index is applied and tested by Di Lorenzo et al. (2020) in unconsolidated aquifers in Italy, which are located in nitrate vulnerable zones. They refined the index (wGHIN) and demonstrated its applicability on shallow and deep aquifers, yet also revealing that the new index has its limitations in terms of low correlations between the indicators."

Methods

Comment #7: Line 116: replace 'with' to 'was'

Response: We agree. Done.

Comment #8: Line 116-117: improve sentence structure 'is mainly caused by' is incorrect, consider 'which' is mainly caused by: : : or rewrite sentence appropriately.

Response: We agree. Done. We replaced 'is mainly caused by' by 'which is mainly caused by'.

Comment #9: Line 120-124: condense and combine sentences

Response: We agree. Done. We condensed and combined sentences in this paragraph as follows: "A contaminant plume, which contains a polycyclic aromatic hydrocarbons concentration of up to 500 $\mu$g/l, of 200 m length over the entire aquifer thickness is located at a former gas plant in the east of Karlsruhe (Figure S1b) (Kühlers et al., 2012). Moreover, three parallel contamination plumes, of 2.5 km length each, can be found in the southeast of Karlsruhe (Figure S1b), where highly volatile chlorinated

HESSD

hydrocarbons (7 $\mu$g/l – 26 $\mu$g/l) and their degradation products were detected (Wickert et al., 2006)."

Comment #10: Line 139-140: belongs in the results section not methods Section

Response: We agree. Done.

Comment #11: 2.3. I found this section hard to read, particularly due to some grammatical errors. This section is too long and verbose, it needs to be rewritten to make it clearer. The second paragraph starts well. I suggest removing the sentence start on line 160 "this requires to obtain also: : :."

Response: We agree on the grammar and the length of this section. Thus, we rewrote the paragraph and deleted some sentences (for more details see marked manuscript, lines 160-167).

Comment #12: Statistical analysis: You do not mention any of the statistical analysis competed in this paper. To be able to distinguish between forest and urban areas, you should at a minimum be completing some statistical analysis of the water quality data you have collected, even if this is simple ANOVA or t-test analysis. This is a major issue that detracts from the quality of this paper. I understand that you have used average values of the sampling wells, however determine whether there are statistical differences between (for instance) temperature at forested areas in comparison with urban areas, and look at the relationships between temperature, well depth and landuses. This analysis would greatly improve the scientific credibility of this study.

Response: We partially agree. We therefore performed U-tests instead of the suggested t-test, due to the possibility that the abiotic data does not follow a normal distribution. The results are as follows:

GWT forest vs. urban area: U = 248, p-value = 3.3×10-5→ significant

Depth vs. GWT (forest): U = 64, p-value = 1.6×10-4 → significant

Depth vs. GWT (urban area): U = 203.5, p-value = $5.5 \times 10^{-5}$ → significant

Hence, we added the p-values of the U-tests to the manuscript (e.g. page 10, line 207). However, as various studies demonstrated (e.g. Amrhein et al., 2019) using p-values alone in a statistical analysis can lead to spurious interpretations, because p-values can exhibit wide sample-to-sample variability and therefore do not reliably indicate the strength of evidence against the null hypothesis (Halsey et al., 2015). Thus, we show the determined p-values, yet our focus remains on the presented spatial analysis using box-plots and other visual tools and comparisons.

Moreover, we added the following introductory sentence in Chapter 2.2. (line 147): "Mann-Whitney-tests (U-tests) were applied to detect potential impacts of groundwater characteristics (physical-chemical parameters), geology and well design on the groundwater quality as well as on groundwater fauna. Samples were regarded as significantly different if the p-value was $<5.0 \times 10^{-2}$."

Results/Discussion

Comment #13: Line 180: complete statistical analysis to indicate if there are significant differences in temperature between urban and forests areas- it appears that there are.

Response: We partially agree and added p-values of U-tests (see previous reply to comment #12).

Comment #14: Lines 192: while the box plots show that there are differences between forests and urban areas in DO and nitrate with landuses, these do not appear to be statistically different. I am not convinced that there are differences in DO and Nitrate between landuses this needs further discussion, as does the large differences in n values between the landuses.

Response: We partially agree (see previous reply to comment #12). The statistical analysis of the content of dissolved oxygen as well as of nitrate in the forest and urban area reveals significantly different distributions, which were added to the manuscript.

Furthermore, we agree that further discussion is needed with regard to the differences between both parameters and land use. Thus, the following paragraph was added in the manuscript (lines 217-221):

"In the urban area average nitrate concentrations are typically higher and correlate with the content of dissolved oxygen (U = 278, p-value = $4.0 \times 10^{-3}$) showing the link between nitrate content and oxygen consumption. Wells with a content of dissolved oxygen below 1.5 mg/l have an average content of nitrate of 1.5 mg/l, caused by nitrate reduction under anoxic conditions. Groundwater with reducing conditions (< 5 mg/l dissolved oxygen) has an average nitrate content of about 7 mg/l in contrast to groundwater with oxidising conditions with 9 mg/l, which is characterised by the oxidation of ammonium to nitrate."

We agree that the results would be more meaningful, if more measurements wells were considered. However, this is beyond the scope of this study and therefore should be part of a future large-scale study.

Comment #15: Line 196: References in chronological order

Response: We agree. Done.

Comment #16: Line 201: 'hold back' should be 'retained' or 'held back' Line 201: suggest these sentences are combined and reduced eg ': : :..where atmospheric nitrogen in retained by forest soils and fertilization is prohibited due to water protection regulations'

Response: We agree. Done.

Comment #17: Line 207-209: Again you cannot claim 'clear differences' without adequate statistical analysis of these factors. You need to run further analysis of the data for this statement.

Response: We partially agree (see previous reply to comment #12). Thus, we added p-values from U-tests to corroborate these observations (see also comment #13 &

[Figure]

**14).**

Comment #18: Line 231: I would not say that amphipods 'predominantly live within wells' rather they have a habitat preference for open spaces such as wells.

Response: We agree. We reformulated the sentence: "Stygobiotic Amphipods, large-bodied invertebrates which due to their size have a habitat preference for open spaces such as wells, . . ."

Comment #19: Paragraph starting line 231: It appears that amphipods are significantly higher in forested areas than urban areas, however without analysis this cannot be determined. This may be ecologically important and should be discussed. It is also worthwhile looking at the correlation between cyclopoida and amphipods as briefly mentioned in line 238.

Response: We partially agree (see previous reply to comment #12). Hence, we performed U-tests and added the corresponding p-values and the following sentences in the manuscript (line 264): "Although statistical analysis with U-tests showed no significant correlation between the abundance of Amphipods and land use (U = 92.5, p-value = $1.5 \times 10^{-1}$), the higher number of individuals in the forest area can support the hypothesis that, as mentioned above, Amphipods indicate healthy groundwater ecosystems, as. . ."

"The lack of a statistically significant correlation might also be related to the low number of wells (n = 8) and individuals (n = 46)."

Moreover, we agree that it is worthwhile looking at the correlation between the abundance of Amphipods and the order Cyclopoida. The following correlations were found and added in the manuscript (line 275). The results of the total statistical analysis are as follows:

Abundance Amphipoda forest vs. urban area: U = 92.5, p-value = $1.5 \times 10^{-1}$ → not significant

Abundance Cyclopoida forest vs. urban area: U = 46.5, p-value = 5.0×10-3 → significant

Abundance Amphipoda vs. Cyclopida (forest): U = 8, p-value = 9.0×10-3 → significant

Abundance Amphipoda vs. Cyclopida (urban area): U = 311, p-value = 2.0×10-3 → significant

Abundance Amphipoda vs. Cyclopida: U = 430, p-value = 9.6×10-5 → significant

Comment #20: Line 238-240: Incorrect grammar: : : remove 'be'

Response: We agree. Done.

Comment #21: Line 248: Incorrect grammar

Response: We agree. We rewrote the sentence as follows:

"In addition, quantities of Bathynellacea (371 individuals) were found in five monitoring wells, all located in the urban area, in a depth of 9.0 to 13.5 m at a GWT of 12-15 °C (Figure 4b)."

Comment #22: Line 274- 280: The issue of purging wells needs further discussion as this is a limitation of your study. If you are looking at proportions of crustaceans to oligochaetes this is almost certainly affected by sampling method. The sentence on line 277 needs to indicate that relative abundances and proportions of crustaceans is likely to be impacted by the sampling methods, thus caution must be taken when interpreting the results.

Response: We partially agree that the sampling method is a limitation of our study. The standing water in the monitoring wells can host a larger number of individuals, caused by filtration effects. Yet, the proportional differences between the two groups are similar between wells and aquifers, as already demonstrated by various previous studies (Hahn and Gutjahr, 2014; Hahn and Matzke, 2005; Korbel et al., 2017). Moreover, only large Amphipods, which were found in three wells, prefer living in open space (well

water) (Hahn and Matzke, 2005; Korbel et al., 2017). Other Crustaceans are smaller, like the order Cylopoida, and are not influenced by filtration effects.

We do not agree that the sentence in line 277 needs to indicate that relative abundances and proportions of crustaceans is likely to be impacted by the sampling methods, as justified by the explanation above.

However, we added the following sentence (line 329): "Nevertheless, pumping can result in the selection of the taxa, especially in the presence of very fine sediments, and can result in changes of the sediment composition in the surrounding of wells and therefore in changes of habitat conditions."

Thus, we are confident about the appropriateness of the used sampling method, yet also mention that caution must be taken when interpreting the results (line 323).

Comment #23: Lines 295 -300: Could this also be due to organic carbon supply? Would level 2 assessment clarify these issues if it were undertaken?

Response: We agree that the food supply is one of the most limiting parameters for the survival of groundwater fauna. Thus, we added the following sentence in the manuscript (lines 355-357): "If the organic carbon supply varies on a small scale, this can influence microbiology and therefore groundwater fauna as well, although short-term changes in nutrient supply can be compensated by groundwater fauna."

We agree that the application of Level 2 might help to get a better understanding of the living conditions of groundwater fauna and might explain why some measurement wells are not populated. The assimilable organic carbon is one indicator which can be chosen as criteria (of the category microbiology) for the evaluation according to Level 2. However, the application of Level 2 is time-consuming and cost-intensive.

Comment #24: Line 305: Could the high (35%) of urban areas displaying natural sites be due to the sample methodology; ie were they classified as good incorrectly due to high proportions of crustaceans that may be influenced by the lack of purging of the

wells?

Response: We partially agree. The standing water of the monitoring wells can contain a larger numbers of individuals than the surrounding aquifer, because wells serve as traps for the groundwater fauna and filtration effects can occur. Yet, as mentioned above, the proportional difference between the two groups will be similar, which is the main criterion for good ecological conditions in this assessment. In addition, some smaller Crustacean (e.g. of the order Cylopoida), which were found in larger numbers in most wells, are not influenced by such effects.

Conclusion

Comment #25: The conclusions of this study to me indicate that the method you have adopted (ie net sample wells and use the proportions of oligo/crustacean populations to determine ecosystem condition) need to be investigated further. The disproportionate number of crustaceans in wells due to sampling methods may be impacting the assigning of "OK" condition to sites that are actually impacted. Potentially a wider range of indicators need to be used including expanding on the use of only oxygen concentration in the classification scheme. The Level 2 assessment (Figure 1) also needs to be discussed in the conclusion.

Response: We disagree that measurement wells are incorrectly classified as good due to the sampling method (see comment #24). This study is focusing on existing approaches to obtain an initial impression on this complex topic. However, we fully agree that there has to be a defined sampling method to achieve representative sampling and comparable assessment of groundwater fauna in the future (see line 336).

We agree that a wider range of indicators has to be used in such classification schemes. Thus, we added the following sentences to the conclusion (line 386): "Level 2 assessment of Griebler et al. (2014) can help to achieve a more reliable and quantitative ecological assessment of urban aquifers, as it divides groundwater ecosystems in ecological grades according to the intensity of anthropogenic disturbance. It is

based on the use of local reference values and the collaboration with experts, which is however challenging to apply. Therefore, further studies with large-scale and repeated measurement campaigns are needed to verify our findings. This should also include other cities and the determination of undisturbed local reference values which are required for a more reliable and also quantitative ecological assessment of urban aquifers. Moreover, a wider range of indicators should be considered in a classification scheme, such as temperature, porosity of the aquifer, groundwater flow, pollutants, nutrient supply, etc., especially when investigating urban areas. In addition, an important adaptation for an improved evaluation method is the determination of fauna at species level, which will provide more information (i.e. about Stygobionts, Stygophiles, Stygoxenes) and also consider the endemism of stygobiotic species. In this context, classification schemes should pay more attention to the different groundwater species and their potential use as indicator species."

References

Amrhein, V., Greenland, S. and Mchhane, B.: Retire statistical significance, Nature, 567, 305–307 [online] Available from: https://www.nature.com/articles/d41586-019-00857-9, 2019.

Brielmann, H., Griebler, C., Schmidt, S. I., Michel, R. and Lueders, T.: Effects of thermal energy discharge on shallow groundwater ecosystems, FEMS Microbiol. Ecol., 68(3), 273–286, doi:10.1111/j.1574-6941.2009.00674.x, 2009.

Griebler, C., Stein, H., Hahn, H. J., Steube, C., Kellemrann, C., Fuchs, A., Berkhoff, S. and Brielmann, H.: Entwicklung biologischer Bewertungsmethoden und -kriterien für Grundwasserökosysteme, Umweltbundesamt., 2014.

Hahn, H. J. and Fuchs, A.: Distribution patterns of groundwater communities across aquifer types in south-western Germany, Freshw. Biol., 54(4), 848–860, doi:10.1111/j.1365-2427.2008.02132.x, 2009.

Hahn, H. J. and Gutjahr, S.: Bioindikation im Grundwasser funktioniert – Erwiderung zum Kommentar von T. Scheytt zum Beitrag "Grundwasserfauna als Indikator für komplexe hydrogeologische Verhältnisse am westlichen Kaiserstuhl" von Gutjahr, S., Bork, J. & Hahn, H.J. in Grundwasser 18 , Grundwasser, 19(3), 215–218, doi:10.1007/s00767-014-0266-4, 2014.

Hahn, H. J. and Matzke, D.: A comparison of stygofauna communities inside and outside groundwater bores, Limologica, 35, 31–44, 2005.

Halsey, L. G., Curran-Everett, D., Vowler, S. L. and Drummond, G. B.: The fickle P value generates irreproducible results, Nat. Methods, 12(3), 179–185, doi:10.1038/nmeth.3288, 2015.

Korbel, K., Chariton, A., Stephenson, S., Greenfield, P. and Hose, G. C.: Wells provide a distorted view of life in the aquifer: Implications for sampling, monitoring and assessment of groundwater ecosystems, Sci. Rep., 7(July 2016), 1–14, doi:10.1038/srep40702, 2017.

Kühlers, D., Maier, M. and Roth, K.: Sanierung im Verborgenen, TerraTech Sanierungspraxis, 3, 14–16, 2012.

di Lorenzo, T., Fiasca, B., di Camillo Tabilio, A., Murolo, A., di Cicco, M. and Galassi, D. M. P.: The weighted Groundwater Health Index (wGHI) by Korbel and Hose (2017) in European groundwater bodies in nitrate vulnerable zones, Ecol. Indic., 116 [online] Available from: https://doi.org/10.1016/j.ecolind.2020.106525, 2020.

Spengler, C.: Die Auswirkungen von anthropogenen Temperaturerhöhungen auf die Crustaceagemeinschaften im Grundwasser, Universität Koblenz-Landau., 2017.

Wickert, F., Muller, A., Schäfer, W. and Tiehm, A.: Vergleich hochauflösender Grundwasserprobennahmeverfahren zur Charakterisierung der vertikalen LCK-W‐Verteilung im Grundwasserleiter, Altlastenspektrum, 01, 29–35, 2006.

[Figure]

Please also note the supplement to this comment:
https://hess.copernicus.org/preprints/hess-2020-151/hess-2020-151-AC1-supplement.pdf

―――――――――――――――――――

---

## Author Comment (AC2) · 30 Jul 2020

Dear referee,

we would like to thank you for your time and the constructive comments, which helped to improve the quality of the manuscript. Please find our detailed replies on the comments below. We hope that we answer all your remarks.

Referee #2:

This study on the distribution of groundwater fauna in the shallow subsurface of urban (city of Karlsruhe) and rural (nearby forest) areas, as well as the use of groundwater fauna for the assessment of the ecological status in groundwater has considerable scientific novelty. To my knowledge, this is the first study that investigates groundwater

ecological aspects in a city's subsurface. This strength by novelty, however, is kind of counteracted by serious weaknesses. While I like the study very much on one hand, it is a pity that the authors did not spend enough time to distill the best out of it. Besides obvious shortcomings in the study design (the selection of chemical parameters measured, the restriction to only well water), the authors did not dig at all into the data available in a 'statistical' sense. I do see more categories of land-use types. I think measures such as well depth and origin of groundwater (what if groundwater impacted in the urban area travels underneath the forest where it is sampled) would be interesting aspects to evaluate. Moreover, fauna data set comes along with further information that has not been used, i.e. the Shannon-Wiener biodiversity and the ratio of stygobites/stygophiles vs. stygoxenes. Not to talk about the determination of individual 'species' of Crustaceans and other groups of animals that could resolve the picture much more. Although, the basic water chemistry in the urban groundwater exhibits some differences to the groundwater sampled in the rural area, there is obviously no clear indication for a 'contamination' of the urban groundwater. An exception is only some temperature deviations. Thus, why it is expected that the groundwater fauna in the urban area is different. I would have loved to see a few hypotheses that are tested. When reading the preprint I also got the impression that most groups of groundwater fauna are described as quite temperature tolerant, however, other publications of the same authors claim the strong sensitivity of groundwater fauna upon groundwater warming. I really missed individual statistical testing of such questions. To be very honest, the paper addresses a really interesting topic that ghas ahrdly been studied to date but has not been properly prepared before submission. My feeling is also that some of the co-authors have not spend much time with the paper, otherwise it would not contain so many flaws. In the following, I will try to provide detailed comments that may help to improve the manuscript. Overall, I not sure if the paper, even when reworked properly, will satisfies the high standard of HESS.

Response: Thank you for the critical assessment of our study. We address your specific comments in more detail below.

Specific comments

Comment #1: P1 L19-20: How have the anthropogenic impacts be measured. I agree that elevated temperature may be seen as Impact. What else? The groundwater chemical analyses do not focus on any contaminants, with exception of nitrate; and nitrate concentrations are not elevated.

Response: We agree that our study is focusing on temperature and nitrate concentrations as important anthropogenic impacts on groundwater ecosystems. We now specifically mention these two proxies in the abstract and also cited the publication by Griebler et al. (2016) in the Introduction (see lines 76-78).

We are aware that there are more potential anthropogenic impacts, such as contaminants, which are not specifically considered in this study. We further agree that more investigations are necessary in the future, as there are likely to be more influencing factors on groundwater fauna distribution such as the sediment, groundwater flow, pollutants, nutrient supply, well design, etc. Thus, we added further research in the supplement of the manuscript and a short summary about what the urban impact is in the summary of chapter 3.1 (see comment #26 and comment #25 Referee #1).

Comment #2: P1 L21: it is mentioned here that more comprehensive assessment methods are required to fully capture the different effects on groundwater fauna. I agree. However, you should mention, at least in the discussion section, what you think of.

Response: We agree that potential strategies for more comprehensive assessment methods should be discussed in more detail. Hence, we added some information to the manuscript, which is also presented in the reply to comment #25 by Referee #1 and in the manuscript.

Comment #3: P2 L44-50: This paragraph does not seem to be linked to the what is introduced before and after.

Response: We agree. Hence, we linked the paragraph by adding the following introductory sentence (lines 44-45): "The availability of ecological criteria can only be increased by conducting a large number of studies dealing with the analyses of groundwater ecosystem health by investigating groundwater fauna."

Comment #4: P2 L51: There is a pile of studies dealing exactly with that. You should name some as examples. What is really new with your study is that there is hardly anything investigated in urban areas.

Response: We agree. We reformulated the sentence and added some studies as follows: "Accordingly, stygobiotic biodiversity is still likely to be underestimated, although there are various studies on this topic (e.g. Gibert and Deharveng, 2002; Malard et al., 2002; Deharveng et al., 2009; Dole-Olivier et al., 2009a)."

Comment #5: P2 L53: Are you sure that the European Union (FP5) PASCALIS project focused on 7 North-American regions. Please check that again.

Response: We agree that the PASCALIS project is focusing on six European regions only. Gibert et al. (2009) focus on six European, as well as seven North-American regions. Thus, we deleted this part of the sentence.

Comment #5: P3 L57: If you state here that regional features have a stronger influence on groundwater fauna than local habitat features, you should test that with your data set. If this is true, maybe the anthropogenic impacts are not strong enough to overrule the regional selective forces. This point should also be discussed.

Response: We agree that this point should be tested with our data and discussed. Thus, we added the following information and results from an additional statistical analysis to this paragraph (lines 307-322), dealing with local habitat features:

"One important natural influence is the local geology, as fine sands and silts are typically rather harsh environments, resulting in an impoverishment of specific groundwater fauna such as Crustacea (Hahn, 1996). The city of Karlsruhe is located on carbonate

('Würm') gravel and river terrace sands, pervaded by bands of drifting sand and inland dune sands. These sediments are highly water-permeable and show almost exclusively vertical seepage water movement. Flood sediments (on top of the river gravel) and bog formations are located in the east and west of Karlsruhe (Regierungspräsidium Freiburg, 2019). This local geological limits the cavity size and therefore impacts the habitat of the groundwater fauna (Wirsing and Luz, 2007). For example, individuals of the genus Parastenocaris typically inhabit small-scale cavity systems (Spengler, 2017). Individuals of this genus can be found both in the wells drilled in gravel (4 wells) and in drifting sand sediments (3 wells) (abundance Parastenocaris vs. geological units: U = 216, p-value = $1.4 \times 10^{-9}$). Amphipods are predominantly to be found in measurements wells located in the 'Würm' gravels (in 5 of 7 wells) (abundance Amphipoda vs. geological units: U = 180.5, p-value = $9.0 \times 10^{-11}$). Moreover, it seems that differences in the geological units have an influence on the total amount of individuals (U = 1312.5, p-value = $1.7 \times 10^{-9}$) and the relative amount of detritus (U = 476, p-value = $3.0 \times 10^{-3}$). As these results show, regional geology has an influence on the occurrence of individual groundwater species and on the amount of individuals as well as on food supply. However, it is not possible to give a reliable estimate of the strength of the anthropogenic impacts, e.g. if they are strong enough to overrule the regional selective forces. Hence, this should be investigated in more detail in future studies."

Comment #6: P3 L83: you could also have used a different approach to look at your results. What if you treat the forest samples as your local natural reference? Just an idea. Starting from there, you could evaluate which well downtown Karlsruhe match natural conditions and which not. Currently, you obviously use a German-wide reference conditions and thresholds (Crustaceans >50%, worms <20%) that may not 'absolutely' reflect the situation in the natural surroundings of Karlsruhe.

Response: We fully agree that it might be useful to define local thresholds by using the forest samples. Thus, we tested the proposed idea. We used the average values of all wells in the forest to define local 'natural' conditions. The calculated values are as

follows:

Chemical: 8 mg/l O2, Nitrate: 2.7 mg/l

Physical: 10.7 °C, a high content of detritus

Faunistic: 28 % Oligochaetes, 66 % Crustaceans, 6 % other individuals, in average three taxa in each well, 131 individuals in each well, average Shannon Diversity Index: 0.7

The biggest issue with these conditions arises from the temperature (no well in the urban area is as cold as the wells in the forest) and the low nitrate concentrations in the forest. Without consideration of temperature and nitrate content, four wells in the urban area are in accordance with the new 'natural' reference values. As this number is similar to the original approach with the German-wide reference, it appears that these new thresholds cannot reflect the complex situation in the urban area of Karlsruhe. Thus, we decided not to adopt this idea for our paper.

Comment #7: In P3 L86: you say that the authors of the UBA study come to the conclusion that aquifer typology is more important than local features. Is this what you say? Why this is not properly discussed in your paper?

Response: We agree that this discussion is missing. Hence, we added a corresponding paragraph in the manuscript (lines 307-322, see comment #5 above).

Comment #8: P4 L113: If 56% of the city's area is covered by vegetation, doesn't it make sense to group the wells in the urban area according to their 'land type' on top and do some statistical analyses?

Response: We disagree to subdivide the wells in the urban area according to their land use due to the following reasons. The total city area of Karlsruhe contains not only the inner city centre and the neighbouring districts (termed "urban area" in this study), but also parts of the Hardtwald and several less built-up outskirts, which results in the high proportion of vegetation in the official 'city area'. The urban area itself however does

not contain enough green spaces to justify a more detailed subdivision and statistical analysis. In our opinion, there is a risk of over-interpreting the results by following this approach. Instead, sampling of groundwater fauna and parameter measurements should be repeated before more emphasis is put on other influencing factors, such as land use. To clarify the issue, we added the information about the definition of the city area in the manuscript (lines 117-118): "Based on the land use plan of Karlsruhe, about 20 % of the area of Karlsruhe (i.e. urban area, city centre, neighbouring districts, as well as parts of the Hardtwald forest and several outskirts), is covered by buildings."

Comment #9: P5 L127: It is mentioned here that the sampling took place between 2011 and 2014 and 39 wells have been sampled. But how often each well were sampled is not mentioned. Did I miss it. 3 times, as said in line 134? Or more often? You cannot have followed the recommendations of Hahn & Gutjahr published in 2014 when sampling took place between 2011 and 2014.

Response: We agree that this information is not mentioned explicitly, but would help to understand the framework of this study. We added this information in the part 'Material and sampling' (lines 141-143) and in Table S1 in the Supplements: "Every well is sampled at least three times. From 2011-2012, 22 measuring wells (mainly in the Hardtwald and the North-West of Karlsruhe) were sampled six times at an interval time of at least two months. In 2014, 17 measurement wells, mainly located in the south/inner city, were sampled three times."

Thus, our approach is in agreement with the recommendations given by Hahn and Gutjahr (2014).

Comment #10: P5 L134: what you mean with 'integrative sampling'? Explain!

Response: By "integrative sampling" we mean taking multiple samples repeatedly over a period of time. We agree that this explanation should be in the manuscript and therefore added this information (line 139).

Comment #11: P5 L136: replace 'groundwater ecology' by 'groundwater ecological status' ; we 'sampled' the fauna. . .

Response: We agree. Done.

Comment #12: P6 L145: If this table shall stay in the paper then the information provided with the individual groups of organisms asks for a balancing. The provided information is very heterogeneous. Some of the terms used have not been explained before, e.g. 'stygophile'.

Response: We agree that some terms are not explained before their use. Thus, we edited Table 1 and added a footnote, which explains the term 'stygophile' (see manuscript).

Comment #13: P7 L156: No, I do not agree at all. There is many natural groundwaters in good ecological shape that do not contain any dissolved oxygen, they may also produce ochre where they come in contact with oxygen. I guess we agree that these sites are not 'good' habitats for groundwater fauna. However, the absence of fauna does not necessarily mean a disturbed ecosystem status.

Response: We agree that the absence of fauna does not necessarily mean a disturbed ecosystem status. In our opinion, it is necessary to clarify that the ecological assessment takes place on the basis of groundwater fauna. Thus, we added the following sentences to Chapter 2.3 and edited the caption of Figure 1 in the manuscript: ". . .If an ecological assessment of groundwater ecosystems on the basis of the groundwater fauna takes place, some faunistic criteria have to be considered. Invertebrates avoid habitats that are ochred or have a low content of dissolved oxygen. Thus, unstressed or natural habitats are defined as. . .".

Comment #14: P7 L170: Doesn't it make sense to further categorize the land use types, also within the city limits?

Response: We disagree. In our opinion, a further subdivision is not reasonable. The

aim of this study is to give a first overview of the ecological groundwater conditions of the study area, so we decided to use only these two major categories (see also reply to the previous comment #8). Also, a subdivision of the urban area into "inner city" and "north-western city", including industrial areas, in an earlier version of the study lead to similar results.

Comment #15: P9 L177: did you also consider well depth in your data analysis. It si a big difference between 8.5 and 39m below land surface which may affect occurrence of fauna and the availability of dissolved oxygen.

Response: We agree that the depth of wells can have an impact on the availability of dissolved oxygen and the occurrence of fauna. In our study, only two wells have a depth of over 16 m (in detail: 27 m and 39 m). The deepest well is uninhabited and has a content of dissolved oxygen of 0.97 mg/l. As the statistical analysis shows, the correlation between well depth and the total amount of individuals is not significant (U = 622.5, p-value = $1.7 \times 10{\text -}1$), but there exists a correlation between the depth and content of dissolved oxygen in the wells (U = 1,478.5, p-value = <10-13). Thus, we added the following sentences in the Chapter 'Physical and chemical parameters' (lines 209-213):

"Moreover, it seems that with a greater depth of the measurement wells the content of dissolved oxygen is increasing (U = 1,478.5, p-value = <10-13). This can be explained by the fact that shallow wells can have a low water column, in which oxygen can rapidly be consumed by groundwater microorganisms, chemical reactions and/or groundwater fauna. In the upper unscreened part of deeper wells, dissolved oxygen can be consumed, while in the screened lower part oxygen is continuously refilled by oxic groundwater from the surrounding (Malard et al., 2002)."

This results shows that the content of dissolved oxygen depends on depth, but depth has no direct influence on groundwater fauna in this study.

Comment #16: P10 Figure3: why are there two lines (red and blue) indicating the

percentage of wells with good and affected ecological status? Automaticly one looks if the box is above or below. However, the values of the individual physicaö-chemical parameters are not in line with the ecological status. I recommend to delete the lines.

Response: We tend to agree. However, we would like to keep both lines and the important information given by these lines. Hence, we reformulated the caption of Figure 3 as follows: "Boxplots of the physical and chemical parameters for the forest and urban area in the study site and the proportion of wells in which ecological conditions are O.K. in percentage [%] indicated by the blue (forest area) and red (urban area) lines (secondary axis);. . ."

Comment #17: P10 L192: To my understanding, a concentration of 1 mg/L dissolved oxygen in wells water strongly indicates that there are anoxic conditions in groundwater. As is mentioned I the preprint well water is not representative for groundwater. To my opinion, since well water is open to the atmosphere, DO concentrations are likely to be overestimated. Gw fauna may, at times of elevated DO in groundwater migrate through the local subsurface and enter wells. There, they may outlast times of no oxygen in the surrounding aquifer. Frankly speaking, I am not sure if the threshold of 1mg/L of DO mentioned before should refer to the surrounding groundwater.

Response: We partially agree that the content of dissolved oxygen (DO) in the well could differ from the content in the surrounding groundwater. In our study, the water and the groundwater fauna are sampled in the well swamp. In the upper unscreened part of the well, DO can be consumed. The lower screened part of the well can be continuously refilled with oxic or anoxic groundwater of the surrounding (Malard et al., 2002). In addition, in the study by Hahn and Matzke (2005) and Korbel et al. (2017) hydro-chemical data such as temperature, pH and DO of the sampled well water and the surrounding groundwater shows no significant differences.

For these reasons, we assume that the content of DO, as wells as the threshold of 1 mg/l, of sampled well water in our study is also representative for the aquifer. Never-

theless, to clarify this issue we already added a paragraph (see comment #15) and the information in brackets '(temperature, pH, dissolved oxygen, etc.)' in the corresponding paragraph (Chapter 3.2 in the end).

Comment #18: P10 L198: The study does not show high nitrate concentration! When stating that $\leq$ 10 mg/L is natural, then in consequence nitrate concentrations between 1.3 and 14mg/l are not high!

Response: We agree. We therefore replaced 'high' by 'higher'.

Comment #19: P10 L199: In general, their relationship between DO and nitrate is not inversely correlated. Only when the oxygen is gone nitrate is reduced. As such, low or no oxygen goes along with low or no nitrate. I do not get the 'link' between oxygen and pollution claimed here (P10 L200).

Response: We agree that the relationship is not inversely correlated in general, and that the link between oxygen concentrations and pollution is not explained. Thus, we added the following sentences to the manuscript: "In the urban area average nitrate concentrations are generally higher and correlate with the content of dissolved oxygen (U = 278, p-value = $4.0 \times 10^{-3}$) showing the link between nitrate content and oxygen consumption. Wells with a content of dissolved oxygen below 1.5 mg/l have an average content of nitrate of 1.5 mg/l, caused by nitrate reduction under anoxic conditions. Groundwater with reducing conditions (< 5 mg/l dissolved oxygen) has an average nitrate content of about 7 mg/l in contrast to groundwater with oxidising conditions with 9 mg/l, which is characterised by the oxidation of ammonium to nitrate."

Moreover, we added the p-values (dissolved oxygen vs. nitrate concentration) for the forest area in the manuscript (line 217).

Comment #20: P10 L199: P11 L220: does this mean that Parastonocaris and Bathynellacea are 'type'-species (groups) for urban situations? Such a possibility is not discussed in the paper. There are groundwater ecology experts in the list of authors. I miss an in depth interpretation of the ecological data.

Response: We partially agree that an in-depth interpretation of the ecological data is missing in the study. However, making a statement about a type species for urban areas on the basis of a single study area with a limited number of measurement wells does not seem reasonable. For this reason, we only hint at the possibility that these two species might be indicators of disturbed and stressed habitats.

Comment #21: P13 L231: Is it that stygobiont amphipods live predominantly within wells? This is to my opinion not a correct interpretation of what is published in Hahn & Matzke (Hahn is co-author of this preprint) and Korbel et al.

Response: We disagree that the interpretation is incorrect. In our opinion, the studies of Hahn and Matzke (2005) and Korbel et al. (2017) indicate that stygobiotic Amphipods have a habitat preference for open spaces, such as wells, and therefore can be found predominantly within wells. However, to eliminate any misunderstandings and in accordance with comment #18 of Referee # 1, we rewrote this sentence (see comment #18 Referee #1 and manuscript).

Comment #22: P13 L264: When the authors write about 'groundwater quality' it is not straightforward what is meant. Only very basic water chemistry (e.g. selected nutrients, pH, DO) and temp was measured. There is no indication for a 'bad' or 'impacted' groundwater quality (with the exception in temperature), so why should the groundwater fauna show associated distribution patterns.

Response: We agree that it is not straightforward what is meant by groundwater quality here. Therefore, we substitute the word 'groundwater quality' by 'groundwater chemistry' in this sentence. Also, we carefully checked the manuscript and clarified the differentiation between groundwater chemistry (i.e. chemical parameters) and groundwater quality.

Comment #23: P13 L235: Does the study of Brielmann et al. 2011 really state that

amphipods react sensitive to a gw temperature of $11 \pm 5°C$ (which is natural gw temp in central Europe) or do they refer to a change of ambient gw temp by $11°C$? Check that carefully.

Response: We agree that this sentence can be misinterpreted. Hence, we re-checked the literature and edited the sentence (line 264) carefully as follows: "Although, statistical analysis showed no clear correlation between the abundance of Amphipods and land use (U = 92.5, p-value = $1.5 \times 10$-1), the higher number of individuals in the forest area could support the hypothesis that Amphipods indicate healthy groundwater ecosystems, as they react most sensitive to disturbances such as pollutants (Korbel and Hose, 2011) and groundwater temperature. In laboratory experiments with a thermal tank, Brielmann et al. (2011) found that 77 % of the individuals of the studied Amphipods (Niphargus inopinatus) preferred areas with a temperature between 8 and 16 °C. In addition, Spengler (2017) and Issartel et al. (2005) observed maximum temperatures up to 17 °C."

Comment #24: P14 L266: It would help the reader if you indicate the general groundwater direction in one of your maps (Fig. 2). If the groundwater flow direction in the area is north-west, then it is very likely that groundwater originating from the urban area is travelling below the forest. This point should be discussed as well.

Response: We agree. Hence, we indicate the groundwater flow direction in Figure 2. The groundwater flow direction in the study area is north-west towards the river Rhine.

Indeed, there is a certain likelihood that groundwater originating from the urban area can travel below the forest, although the whole area north-east in the forest area is water protection area. Nevertheless, the "Waldstadt" settlement in the north-east of the city might affect groundwater fauna in the forest area. Thus, we looked up measured chemical parameters of wells provided by the continuous monitoring program of the LUBW. One measurement well is located in the "Waldstadt", next to the wells T411 and T412 of this study. This well shows values in the range of the local background or

threshold of the drinking water ordinance of Germany.

Hence, to clarify this issue we added the following sentences (lines 230-231): "Moreover, no impact of groundwater originating from the urban area on the wells in the forest area is observed."

Comment #25: P14 L285: Why only the two criteria (>70% Crustaceans and <20% of oligochaetes were use for the evaluation of the ecological status. There are more criteria mentioned in the UBA report and in the international literature, some of which have been used or even developed by the co-authors, i.e. the Groundwater Fauna Index, the ratio of stygobites/stygophiles vs. stygoxenes, etc. Making use of these additional measures could provide a much clearer picture.

Response: We agree that using additional measures could provide more comprehensive information. Actually, we tested more methods during the preparation of this study, which we now present in the supplement of the manuscript.

The GFI however did not provide any additional information or valuable insights and was therefore excluded. The influence of multiple stressors, such as the pollution of the groundwater through industrial plants etc., and their effects on the governing parameters can bias the GFI. Moreover, under urban areas changes in GWT are caused by anthropogenic heat inputs (Menberg et al., 2013b, 2013a; Benz et al., 2014; Tissen et al., 2018), rather than being related to surface water influences. Hence, the GFI appears to be unsuitable for the assessment of the groundwater fauna in an urban setting. We added this information to the supplement of the manuscript.

We partially agree that use of the ratio of stygobites/stygophiles vs. stygoxenes is useful in the context of this study. We agree that this ratio will provide more information on the endemism of stygobiotic species. Yet, we decided not to use it, because the required determination of the fauna cannot be done by untrained persons, which was in the sense of the UBA project (Level 1). This information is therefore also not added to our manuscript. The same applies to the GHI, where the microbiological analyses

are beyond the purpose of a first tier assessment.

Comment #26: P15 L300-301: 'as expected, this indicates anthropogenically influenced groundwater ecosystems...'. Again, the physical-chemical data provided do not hint at a seriously 'impacted' groundwater quality. The only exception is the temperature. It would have been worth to expand the list of chemical parameters analyzed and include 'contaminants' besides nitrate which is more of an issue in agricultural land. I ask the authors to make clear in the paper 'what exactly the urban impact' is.

Response: We agree that the focus on nitrate and temperature as anthropogenic impacts in this study has to be clarified (see also reply to comment #1).

In order to expand the list of chemical parameters, we conducted further analysis using data provided by a continuous monitoring system. This information is now given in the supplement and a short summary is presented in chapter 3.1 (line 231-238): "Further investigations demonstrated that beside one larger and two smaller contaminated sites (however, still with concentrations below the threshold values, Figure S1), only minor groundwater pollution is documented beneath Karlsruhe (see Supplement). The chemical and physical parameters considered in the long-term monitoring system are within the range of local background and below threshold values of the drinking water ordinance of Germany (see Supplement for more information). In addition, groundwater fauna can usually cope well with short-term changes of chemical-physical parameters (Griebler et al., 2016). Previous studies showed that some species can even benefit from pollutants (Matzke, 2006; Zuurbier et al., 2013). Thus, the main documented impacts on groundwater quality in the study area are related to temperature and oxygen as well as nitrate concentration."

Comment #27: P15 L304-306: This sentence needs an explanation. Why do the results you obtained lead to the 'offer' of using groundwater for heating and cooling? This sentence is not in line with what has been discussed right before.

Response: We agree that this sentence needs further explanation. Thus, we rewrote

the sentence in the manuscript (lines 364-367) as follows: "This observed spatial heterogeneity in ecological conditions and the existing heat anomalies in the urban area also call for an adapted usage for shallow geothermal energy systems. Areas with no or only little groundwater fauna (i.e. affected habitats) could also be used to store thermal energy at higher temperatures. Thus, high-temperature aquifer thermal energy storage (HT-ATES) could be established in urban environments (e.g. Fleuchaus et al., 2018), where the demand is high."

Comment #28: Discussion section in general: I miss proposals for improvement, i.e. the use of additional parameters, more sampling, more wells, other sampling techniques (here I could find 1 sentence), . . .

Response: We agree. Done (see comment #1, #22 and #25 of Referee #1, see manuscript).

Technical comments

Comment #29: P1 L12: 'scarce' not 'scare'

Response: We agree. Done.

Comment #30: P1 L15: If I am correct, then the classification is from German Federal Environment Agency (UBA) but the result from a UBA funded research project. This makes an important difference. The funding agency not necessarily identifies itself with the outcome of funded projects. The 'invention' and 'responsibility' is with the authors from the study. As such, I would not call the scheme used, and UBA classification scheme. Same applies to P2 L31.

Response: We agree. Done. "For classification we apply the scheme of Griebler et al. (2014), on behalf of the Federal Environmental Agency (UBA),. . ."

Comment #31: P1 L16: wrong wording: 'fine' ecological conditions. Replace by 'good', 'natural' or something similar. Best you use the terminology used with the assessment scheme you used.

Response: We agree. Done.

Comment #32: P2 L26: HESS is an international journal. I would cite 'German' and 'grey' literature only if there is not similar publication in international journals. This is my very personal opinion.

Response: We agree. However, as the study site is located in Germany, it is sometimes necessary to cite 'German' literature, e.g. to get data on regional geology, which is often not available in the international literature. We carefully assessed the cited literature again and found that the mentioned study cannot be replaced and is thus kept in the manuscript.

Comment #33: P2 L28: 'retention' is the wrong term here! What you mean is 'degradation' or 'mineralization'.

Response: We disagree. We mean retention of organic matter by groundwater ecosystems, which react like a buffer and storage zones.

Comment #34: P2 L30: delete 'valuable'. What do you mean with 'tied'? Reword.

Response: We agree. Done. We mean "to bind", e.g. organic matter, by biological processes/microbial activity. For a better understanding 'tied' is replaced by 'bound'.

Comment #35: P2 L34: change 'relatively' to 'typically' or 'naturally'. Typo: Brielmann et al. 2011 not 20011.

Response: We agree. Done.

Comment #36: P2 L35: change âAËŸŽstygobiote' to âAËŸŽstygobite' or 'stygobiont'.

Response: We agree. Done, 'stygobiote' is replaced by 'stygobiont'

Comment #37: P2 L38: . . . groundwater . . . is not yet recognized as a protected habitat . . . reword this part of the sentence. Wat you probably mean is that gw is not yet recognized as an ecosystem that deserves protection.

Response: We agree. Done.

"Nevertheless, groundwater is not yet recognized as a habitat, which is worthy of protection, in German and European legislations."

Comment #38: P2 L39: change 'assessing groundwater ecology' into ' assessing groundwater ecological status'.

Response: We agree. Done.

Comment #39: P3 L59: delete 'Unfortunately'. Not needed.

Response: We agree. Done.

Comment #40: P3 L65: do the temp fluctuations range between 4°C and 20°C or is there a temp fluctuation with a temp range between 4°C and 20°C. Try to be more precise with your wording.

Response: We agree. We have rewritten the sentence: "According to Brielmann et al. (2011) annual temperature fluctuations in aquifers, caused by shallow geothermal energy systems, range between under 4 °C in winter and up to 20 °C in summer."

Comment #41: P3 L73: change 'clearly increasing' to 'increased and 'usually decreases' to 'decreased'.

Response: We agree. Done.

Comment #42: P3 L75: Brielmann et al. 2011 not 20011!

Response: We agree. Done.

Comment #43: P3 L79: The UBA did not develop anything! The UBA funded a research project in which these tools you refer to were developed. Rephrase this sentence.

Response: We agree. Hence, we rephrased this sentence: "Commissioned by the Federal Environmental Agency of Germany (Umweltbundesamt, UBA), Griebler et al.

(2014) developed a concept for an ecologically based assessment scheme for groundwater ecosystems."

Comment #44: P4 L96: change 'waterside filtration' to 'river bank filtration'.

Response: We agree. Done.

Comment #45: P4 L98: 'beneath' an urban area

Response: We agree. Done.

Comment #46: P4 L99: you can not sample thermal properties. You collected or sampled gw fauna and 'analyzed' gw chemistry and measured gw temp.

Response: We agree. Thus, we rewrote the sentence: "Hence, in 39 groundwater monitoring wells in Karlsruhe, Germany, the groundwater fauna is sampled, groundwater temperatures are measured and chemical properties are analysed."

Comment #47: P4 L100: Again, it is not the classification scheme of the UBA.

Response: We agree. Thus, we rewrote the sentence: "In our study the classification scheme developed by Griebler et al. (2014) is applied."

Comment #48: P4 L101: 'state of ecosystem quality' sounds weird.

Response: We agree. Thus, we replaced 'state of their ecosystem quality' by 'the quality of their ecosystem'

Comment #49: P4 L116-117: annual mean LST! Is this what you mean?

Response: We agree. We mean annual mean land surface temperature (LST). Thus, we added 'annual mean'.

Comment #50: P4 L120: Didn't you specifically 'analyze' statistically if wells in the area of known contaminations show different features than others?

Response: We partially agree. Wells in the area of known contaminations can indeed

show different features than others, yet in this study only two measurement wells are close to a known contamination, which makes a statistical analysis infeasible. Hence, we added some information in the manuscript (see our reply to comment #26).

Comment #51: P7 L149: . . .classification scheme in the framework of a research project funded by the. . .

Response: We agree. Done.

Comment #52: P7 L152: O.K. is an improper term in this connection

Response: We agree, but we would like to keep the original phrase and meaning of the original document by Griebler et al. (2014).

Comment #53: P11 L207: chemical characteristics do not distribute! There is distribution patterns.

Response: We agree. Done.

Comment #54: P13 L242: 'Larger'!

Response: We agree. Done.

Comment #55: P13 L252: 8.3 mg/l is 'not' a rather high nitrate content! Same applies to P13 L257.

Response: We agree. Thus, we reformulated the sentence:

". . .and a rather high nitrate content (8.3 mg/l) compared to the wells in the forest area. . ."

[revised manuscript text omitted]

Please also note the supplement to this comment:
https://hess.copernicus.org/preprints/hess-2020-151/hess-2020-151-AC2-
supplement.pdf

———————————————————
151, 2020.

---

## Referee Report (RR1)

The paper has improved significantly since the first publication and is a novel manuscript, providing insight into groundwater biota beneath urban areas and the surrounding landscape. The work is very interesting, however more time needs to be invested in correcting issues highlight below to meet the high standards of HESS.

1. I have difficulties in establishing the types of landuses, some 'forested' areas appear very close to the urban areas on figure 3c. Further clarification and a statement that distinguishes or helps classify landuses would be helpful
2. Did you consider looking at stygobite vs stygoexene ratios, this may have provided more insight into the biotic differences
3. You mention using detritus as a measure, but there is not mention of methodology, unit of measurement is not included, and you have referred to this inconsistently throughout the results
4. It would be good to get an indication of the flow of GW particularly in the areas where forest and urban areas are close. This would help the statement made in line 233 (see below)
5. Whilst language and grammar have improved since the first version, this still requires a good proof-read to remove grammatical and punctuation errors.

Line 51-53: incorrect grammar. Remove 'etc' and combine sentence on line 53 to above paragraph

Line 104: it would be great to see a hypothesis here…

Line 225: One sentence doesn't make a paragraph, combine with previous paragraph

Line 233: 'no impact of GW originating from the urban areas on the wells in forest areas is observer' how does reader interpret this as we do not know flow direction of aquifer?

Line 238-241: this seems to be in the wrong section. Talk about biota in the below section

Line 244: The 'biotic' communities sounds better

Line 274: Also need to clarify that n=8 in ? forested areas. (this should also be stated in the methods section ie 8 wells in forested areas X wells in urban areas)

Lin 295: missing a comma

Line 306: How did you measure detritus (should be in methodology) and what does (>2) mean? What are the units here?

Line 381-383: mention 31 wells in urban and 8 wells in forested areas

385: I would mention that Ampiphod were much more abundant in forested wells than in urban areas

Line 403: remove the 'etc'

---

## Referee Report (RR2)

**Groundwater fauna in an urban area: natural or affected?**

Fabien Koch[1], Kathrin Menberg[1], Svenja Schweikert[1], Cornelia Spengler[2], Hans Jürgen Hahn[2], Philipp Blum[1]

[1]Institute of Applied Geosciences (AGW), Karlsruhe Institute of Technology (KIT), Kaiserstraße 12, 76131 Karlsruhe, Germany
[2]Faculty of Nature and Environmental Sciences (Working Group: Groundwater Ecology), University Koblenz-Landau, Im Fort 7, 76829 Landau, Germany

*Correspondence to*: Fabien Koch (fabien.koch@kit.edu)

**Abstract.** In Germany 70 % of the drinking water demand is met by groundwater, whose quality is the product of multiple physical-chemical and biological processes. As healthy groundwater ecosystems help to provide clean drinking water, it is necessary to assess their ecological conditions. This is particularly true for densely populated, urban areas, where faunistic groundwater investigations are still scarce. The aim of this study is therefore to provide a first  assessment of the groundwater fauna in an urban area. Thus, we  the ecological  of an anthropogenically influenced aquifer by analysing  fauna in 39 groundwater monitoring wells in Karlsruhe (Germany) . For classification, we apply , in which a threshold of more than 70 % of Crustaceans and of less than 20 % of Oligochaetes serves as an indication for good ecological conditions.   revealed that only 35 % of the wells in the  area, and 50% of wells in the  fulfil these criteria.   spatial pattern with respect to land use and other anthropogenic impacts, in particular, groundwater temperature here are  differences in the spatial distribution of species  abiotic groundwater characteristics , which indicates that more comprehensive assessment  required to .

[revised manuscript text omitted]

---

## Author Response (AR2)

**Author's response to editor's and referee's comments on hess-2020-151**

**"Groundwater fauna in an urban area: natural or affected?"**

5 **Dear Editor,**

We would like to thank you for the opportunity to once more revise our manuscript, for your time and for the constructive comments. We are convinced that we have fully addressed now all comments and substantially improved the manuscript.

In general, our replies to the comments are highlighted in blue.

Best regards,

Fabien Koch, on behalf of all authors

**Editor:**

Comments to the Author:

The reviewers re-evaluated the revised manuscript. One of the reviewers was satisfied with most of the corrections and pointed out some details still needing revision (details see attached document). The second
20 reviewer thoroughly commented on the manuscript pointing out some major weaknesses and even errors in the manuscript (see comments referee #2). Normally, this would justify a rejection of the manuscript at this point; after already having had a round of "major revisions". Still, the referee also pointed out that there is novelty in the dataset and gave good advice on how to improve the manuscript. Therefore, there is the very last chance to thoroughly and substantially revise the manuscript. If you think that all of the
25 comments can be addressed and the manuscript can be substantially improved, a re-submission of a revised version is recommended as I will make clear decision on acceptance or rejection of the manuscript in the next round. Please also have again a look at the comments from the first round of revisions.

**Response:** We agree that the referees gave very helpful comments for improving the manuscript. Hence, we additionally performed a more sophisticated multivariate analysis in form of a dimensionality-reduction method for visualization, which revealed interesting insights into parameter relations and confirmed our previous findings about spatial differentiation. Moreover, we improved the manuscript's language and as suggested we looked again at the comments from the first round of revisions for improving the manuscript.

In detail, we added isohypses in Figure 2c of the manuscript to provide details about local groundwater flow conditions as was recommended in Comment#25 of Referee#2 from the first round. Moreover, the results of the additional multivariate test support the categorization of land use types for which we argued in the response of Comment#12 of Referee#1 and Comment#15 of Referee#2, as well as the hypothesis that the order *Bathynellacea* and the genus *Parastenocaris* are type species for urban situations (Referee#2 Comment#21 Round#1).

**Dear Referee #1,**

we would like to thank you for your time and the constructive comments, which helped to improve the quality of the manuscript. Please find our detailed replies on the comments below. We hope that we
45   answer all your remarks.

In general, our replies to the referee's comments are highlighted in blue. To highlight the nature of our replies we use a traffic light system indicating agreement with the referee marked in green, partial agreement in yellow, and objections in red.

50

Best regards,

Fabien Koch, on behalf of all authors

**Referee #1:**

55

The paper has improved significantly since the first publication and is a novel manuscript, providing insight into groundwater biota beneath urban areas and the surrounding landscape. The work is very interesting, however more time needs to be invested in correcting issues highlight below to meet the high standards of HESS.

60   Response: We fully agree. Thus, we added clarifications and statements in the methodology, which are presented below in our replies to the 'general and specific comments'. Moreover, we removed grammatical and punctuation errors (see 'specific comments').

**General comments**

65

Comment #1: I have difficulties in establishing the types of landuses, some 'forested' areas appear very close to the urban areas on figure 3c. Further clarification and a statement that distinguishes or helps classify landuses would be helpful.

Response: We agree that a clarification is necessary to better understand the classification. Hence, we added the following information to the manuscript (lines 205-209):

"In order to allow a spatially differentiated assessment, the study site is classified in different zones based on land use types provided by the European seamless vector database of the CORINE Land Cover (CLC) inventory (GISAT, 2016). Based on this data the study site is subdivided into:(1) Forest area (local name: Hardtwald) and (2) Urban area containing industrial, commercial and residential areas (Figure 2a). A more detailed subdivision in the urban area did not appear reasonable due to the heterogeneous structure."

Comment #2: Did you consider looking at stygobite vs stygoexene ratios, this may have provided more insight into the biotic differences.

Response: As we already mentioned in in the last round of revision in Comment #26 of Referee #2, we partially agree that use of the ratio of stygobites/stygophiles vs. stygoxenes might be useful in the context of this study. We agree that this ratio could provide more insight into the biotic differences. Yet, we decided not to use it, because the required determination of the fauna is not part of the assessment scheme by Griebler et al. (2014) (Level 1). The information is therefore not added to our manuscript, yet we now mentioned this ratio as an important adaptation for future assessment schemes in the conclusion of the manuscript.

Comment #3: You mention using detritus as a measure, but there is not mention of methodology, unit of measurement is not included, and you have referred to this inconsistently throughout the results.

Response: We agree that information about measurement technique and unit of detritus was not mentioned in the methodology yet. Thus, we added the following paragraph (lines 157-161):

"Furthermore, the relative amount of sediment as an indication of the nutrient availability and the cavity system was measured. Before the fauna sample from the net sampler was passed over a sieve with a mesh size of 74 μm, the sediment is separated and classified in different categories (sand, fine sand, ochre, detritus, silt). It should be noted that the detritus content is not recorded

quantitatively but on the basis of estimated frequency classes. The estimation of the relative amounts of sediment per sample is based on Table S1 in the supplement."

**Table S1. Estimation of the relative amounts of sediment per sample (modified after Hahn, 2006)**

| Scale | Description | Characterisation |
|-------|-------------|------------------|
| 0 | Absent | No sediments in the sampling vessel |
| 1 | Little | Bottom of the sampling vessel (Ø ¼ 7.6 cm) slightly covered by sediment |
| 2 | Much | Bottom of the sampling vessel covered by several millimetres of sediment |
| 3 | Very much | Bottom of the sampling vessel covered by one or more centimetres of sediment |

100 Comment #4: It would be good to get an indication of the flow of GW particularly in the areas where forest and urban areas are close. This would help the statement made in line 233 (see below).

Response: We agree that a more detailed indication of the groundwater flow would help the reader to follow this statement. In addition to the statement about flow velocity (lines 260-261) and the general flow direction already indicated in Figure 2c and Figure S1b, we added a groundwater

105 contour map in Figure 2c providing more details about local groundwater flow conditions.

[Figure]

**c** Ecological condition after Griebler et al. (2014)

● Natural  ○ Evaluation not possible  ← Groundwater flow direction
● Affected  -- Groundwater contour map [m a. s. l.]

**Figure 2: Overview map city area of Karlsruhe: … (c) faunistic evaluation after Griebler et al. (2014) and groundwater contour map in metres above sea level (modified after the local authority real estate of Karlsruhe).**

110 Comment #5: Whilst language and grammar have improved since the first version, this still requires a good proof-read to remove grammatical and punctuation errors.

Response: We agree. Hence, we again thoroughly checked the manuscript to remove grammatical and punctuation errors.

115

120

**Specific comments**

Comment #6: Line 51-53: incorrect grammar. Remove 'etc' and combine sentence on line 53 to above paragraph.

125    Response: We agree. Done.

Comment #7: Line 104: it would be great to see a hypothesis here…

Response: We agree. Hence, we added the following sentence to the paragraph (line 113):

"The objective of this study is to investigate specifically the groundwater fauna beneath an urban
130    area in comparison to a natural forest to determine whether land use has an impact on groundwater organism communities."

Comment #8: Line 225: One sentence doesn't make a paragraph, combine with previous paragraph.

Response: We agree. Done. We combine the sentence with the previous paragraph by adding, i.
135    a. the conjunction 'moreover' at the beginning of the sentence.

Comment #9: Line 233: 'no impact of GW originating from the urban areas on the wells in forest areas is observer' how does reader interpret this as we do not know flow direction of aquifer?

Response: We agree and added further information to this paragraph and a groundwater contour
140    map in Figure 2c:

"Moreover, no impact of groundwater originating from the urban area on the wells in the forest
area is observed, as the groundwater flow direction in Karlsruhe is northwest (see Chapter 2.1 and
Figure 2c)."

145    Comment #10: Line 238-241: this seems to be in the wrong section. Talk about biota in the below section.

Response: We agree and moved the two sentences to the below section (lines 336-338).

Comment #11: Line 244: The 'biotic' communities sounds better.

150          Response: We agree. We replaced the word organism by biotic.

Comment #12: Line 274: Also need to clarify that n=8 in ? forested areas. (this should also be stated in the methods section ie 8 wells in forested areas X wells in urban areas)

          Response: We agree. Thus, we added this information in the brackets in line 300. Moreover, we
155          added the following sentence in the methodology (line 143):

"From 2011 to 2014, samplings of groundwater parameters and fauna were performed in 39 groundwater monitoring wells in Karlsruhe, of which eight wells are in the forest and 31 in the urban area."

160  Comment #13: Line 295: missing a comma

          Response: We agree. Added.

Comment #14: Line 306: How did you measure detritus (should be in methodology) and what does (>2) mean? What are the units here?

165          Response: We agree. Thus, we added this information in the methodology (see Comment #3).

Comment #15: Line 381-383: mention 31 wells in urban and 8 wells in forested areas.

          Response: We agree. Done.

170  Comment #16: Line 385: I would mention that Ampiphod were much more abundant in forested wells than in urban areas.

          Response: We agree. Done. We added the following sentence in the conclusion (lines 475-476):

"Moreover, Amphipods are more abundant in wells in the forest than in urban area."

175  Comment #17: Line 403: remove the 'etc'

          Response: We agree. Done.

**Dear Referee #2**,

we would like to thank you for your time and the constructive comments, which helped to improve the
quality of the manuscript. Please find our detailed replies on the comments below. We hope that we
answer all your remarks.

In general, our replies to the referee's comments are highlighted in blue. To highlight the nature of our
replies we use a traffic light system indicating agreement with the referee marked in green, partial
agreement in yellow, and objections in red.

Best regards,

Fabien Koch, on behalf of all authors

**Referee #2:**

The study of Koch et al. has now been revised and individual sections of the manuscript improved
considerably. And while I still think there is substantial novelty in this data set, there is numerous issues
that would need to be seriously addressed before publication. In fact, the manuscript contains 'scientific
errors' that must be removed and draws conclusions that are not supported by the outcome of the study
(see below). […] I am very sorry to disappoint the authors, after putting efforts in the revision of the
original submission, but the manuscript to my opinion is still far from being ready to be published in
HESS. I recommend another round of major revision.

> Response: Thank you for the critical assessment of our study. We address your specific comments
> in detail below.

**Specific comments**

Comment #1: To my opinion, nitrate values found in urban groundwater are comparably low and to my opinion do not point to a strong contamination. Moreover, there is numerous studies that underline that nitrate at concentrations below 50mg/L does not directly affect groundwater fauna. In consequence, one cannot expect much of an outcome in that respect. Indeed, correlations with nitrate have been shown but through indirect effects in agricultural areas. Since only a few physical-chemical parameters have been measured, and only temperature and land use, that show clear alterations to a 'natural' reference situation, I would put my focus on these two 'impacts'.

Response: We agree to put the focus on temperature and land use as major impacts, and modified the manuscript accordingly. In line 266, we deleted "and nitrate concentration" and in line 332 "and the highest nitrate concentrations (> 6 mg/l)". Furthermore, we added the following sentences to the discussion of Chapter 3.2:

"In case of nitrate, numerous studies underline that nitrate at concentrations below 50 mg/l does not directly affect groundwater fauna (Fakher el Abiari et al., 1998; Mösslacher and Notenboom, 2000; Di Lorenzo and Galassi, 2013; Di Lorenzo et al., 2020). As the highest average nitrate content per well is below 15 mg/l in this study, a direct negative effect of the nitrate concentration on the groundwater fauna is unlikely. Thus, nitrate is only mentioned as one measured parameter and is not discussed as a potential anthropogenic impact in this study."

Comment #2: I fully agree that groundwater fauna is temperature sensitive and in central Europe stygobionts are almost exclusively (with some exceptions) cold stenothermic. I do not agree with the thresholds mentioned in the manuscript and the sources cited. It is stated (P2-L37) that groundwater fauna 'cannot withstand' water temperatures over 16°C (Brielmann et al. 2009) or rather 14°C (Spengler et al. 2017) for an extended period. This is definitely not true. I went back into the cited sources and what is stated there is as follows: Brielmann et al. (2009) says "True groundwater invertebrates (stygobites) are assumed to be cold stenotherm and can hardly persist at water temperatures exceeding 16°C for extended periods of time (T.Weber & S.I. Schmidt, unpublished data)." It says 'hardly' and cites work 'not

published' and the paper is 10 years old. The study itself found that "… faunal abundance showed no relation to impacted groundwater temperatures, but faunal diversity decreased with temperature, possibly emphasizing the sensitivity of individual groundwater invertebrates towards heat discharge." No relationship between temperature and faunal abundance! In Brielmann et al. (2011) it is stated that „Niphargus inopinatus (groundwater amphipod) when allowed to move freely in a temperature gradient preferred a temperature between 8 and 16°C; in 77% of the observations the specimen were found there, but in consequence in 23% of the cases the animals were outside this range. For the isopod Proasellus cavaticus, specimen were in 66% of the observations found between 8 and 16 °C. In Glatzel (1990) a species-specific critical threshold temperature of 19°C is mentioned for Parastenocaris phyllura (harpacticoid copepod) beyond which a significantly higher mortality occurred. A study on groundwater microbes and fauna in local aquifers below basins collecting surface runoff during extreme rain events found that groundwater fauna was almost absent at spots that were impacted by significant temperature dynamics, with maximum temperatures of up to 22°C (Foulquier et al. 2011). Spengler et al. (2017) reports about declining fauna biodiversity at temperatures above 14°C. In fact, there is species found that start to disappear from the communities at higher temperatures while others are still found.

If we summarize all this information, then it is clear that there is a variability in temperature tolerance among groundwater faunal groups and species. No clear threshold at 14°C or 16°C appears proven, more likely individual thresholds are somewhere between 14°C and 18-20°C, based on what has been reported so far. It is really essential to carefully interpret findings from other studies and data published.

Response: We fully agree. We reformulated the paragraph carefully and added more studies as follows (lines 36ff):

"Hence, in Central Europe they are assumed to be cold stenotherm which means that they prefer cold temperatures. A variability in temperature tolerance among groundwater faunal groups and species is reported in various studies, which explains why the use of individual temperature thresholds is more useful to capture different preferences. According to Spengler (2017) faunal diversity is generally declining at a temperature above 14 °C. Various authors reported species specific temperature preferences between 8 and 16 °C (for individuals of the species *Niphargus inopinatus* and *Proasselus cavaticus* (Brielmann et al., 2009, 2011)) and a specific temperature

threshold of up to 19 °C (for *Parastenocaris phyllura* (Glatzel, 1990)). Above these thresholds the mortality of individuals raises until groundwater fauna is almost absent, for example at 22 °C in the study of Foulquier et al. (2011). However, temperature sensitivity is not only an issue at species level, but also for the communities as a whole. Spengler (2017) reported 12 °C to be a temperature threshold value indicated by a shift in community structure for faunal communities of groundwater of the Upper Rhine Valley."

Comment #3: To my very personal opinion there is two ways to publish scientific results and findings. First, to do the minimum necessary. Second, to explore the data best possible. My feeling is, and this was already said in the first round of review, that the data set has not yet been explored and analyzed in a proper way. Although there was substantial criticism from both reviewers because of a lack of statistical analyses, the only change that was done is applying now a simple Withney-Mann-U-Test to all data. That is sad and boring, and to my opinion does not deserve publication in a high ranked journal. Only from the papers cited, the authors could have derived ideas about the application of additional, more sophisticated multivariate tests like PCA, CCA, … Sorry to be so direct.

Response: We agree that the dataset should be explored in the best possible way. Thus, we added an additional, more sophisticated multivariate analysis in form of PHATE analysis. The rationale for the selection as well as the description of the method were added in the methodology, the results of the analysis are described in the new chapter 3.4. Moreover, the detailed results of the PHATE analysis were added to the supplement of the manuscript (Figure S1b, S3 & S4 and Table S45 & S5).

[revised manuscript text omitted]

Comment #4: I like the idea of testing the ecological assessment schemes of Hahn (2006), Griebler et al. (2014) and Korbel & Hose (2017) in an urban setting. However, such an application needs to be done with some care. In the first tier (step) of the scheme described in Griebler et al. (2014) which is somehow similar to what was published by Korbel & Hose (2011), it is recommended to choose five or more criteria with a minimum of 3 biological ones. If criteria are selected that are partly dependent to each other, e.g. proportion of crustaceans and proportion of oligochaetes, then the resolution of the assessment is very low. Surprisingly, although several assessment indices have been considered by the authors (GHI, GESI, GFI), results of none are presented in the paper. Obviously, as I got from the reply to reviewers' comments, things have not worked out as clear as expected. I would have liked to read in the discussion about the 'pitfalls' of the individual assessment schemes. Again, an assessment scheme cannot compensate the lack in use of multiple sensitive criteria. Finally, although, the prerequisite to sample stations more than once is fulfilled, sites that are compared have been sampled in different years, a fact that should at least be discussed.

360     Response: We absolutely agree that results from additional assessments, which are not presented in the manuscript (but in the supplement of the manuscript), and the fact that wells have been sampled in different years should be discussed. Hence, we added the following sentences to the manuscript (lines 378-380 & 409-423):

"Care should also be taken when interpreting faunistic results of sites that were sampled in
365     different years. To improve comparison of the biotic communities, a consistent sampling period of every well is necessary in the future."

"Furthermore, the integration of additional biological criteria might help to improve the results of the assessment according to Griebler et al. (2014), as well as the application of different assessments, such as the similarly structured GHI or wGHI$^N$ (Korbel and Hose, 2017; Di Lorenzo
370     et al., 2020b). Moreover, there are a couple of newly developed indexes, like the D-A-C-Index, which is based on microbiological indicators and shows whether groundwater reserves deviate from natural references (Fillinger et al., 2019), which can be used in the future. As mentioned in the introduction, another way to quantify the relevant ecological conditions in the groundwater is the GFI (Hahn, 2006). During the preparation of this study, the GFI was applied to the data (see
375     Supplement), however, it did not provide any additional information or valuable insights. The influence of multiple stressors, such as the pollution of the groundwater by industrial plants etc., and their effects on the governing parameters are likely to bias the GFI. In general, the GFI seems to be suitable only for unpolluted and anthropogenically undisturbed groundwater with sufficient oxygen concentrations (> 1 mg/l). Moreover, in urban areas changes in GWT are caused by
380     anthropogenic heat inputs (Menberg et al., 2013b, 2013a; Benz et al., 2014; Tissen et al., 2018), rather than being related to surface water influences. Hence, the GFI appears to be unsuitable for the assessment of the groundwater fauna in an urban setting. The same outcome emerges for the Shannon diversity index, which was also tested during the preparation of the study and showed no clear distribution pattern according to faunal diversity."

385

Comment #5: I guess, we all agree that this first study of groundwater fauna and assessment of the groundwater ecological status in an urban setting was accompanied by some limitations. There have been

only a few physical-chemical parameters measured, the number of wells sampled werde very different for the two land use categories, and regional and local reference conditions for the groundwater fauna were missing, to give just three examples. This is normal, and one can nicely build on this first experience. And yes, the reply of the authors to several of the reviewer recommendations was: "The aim of this study was to provide a first overview of the ecological groundwater conditions of the study area". What I really disliked is that although the results are of limited validity and transferability, and need to be confirmed in follow-up investigations, at the end of the discussion section it is stated that: "Areas with no or little groundwater fauna could be used for to store thermal energy at higher temperatures." and "HT-ATES could be established in urban environments." How can this conclusion be drawn from the findings presented?

Response: We agree that this conclusion cannot be drawn from the findings anymore. Thus, we removed this paragraph.

**References**

[revised manuscript text omitted]

---

## Author Response (AR3)

**Author's response to editor's and referee's comments on hess-2020-151**

**"Groundwater fauna in an urban area: natural or affected?"**

5 **Dear Editor,**

We would like to thank you for the opportunity to once more revise our manuscript, for your time and for the constructive comments. We are convinced that we have fully addressed now all comments and substantially improved the manuscript.

10

In general, our replies to the comments are highlighted in blue.

Best regards,

Fabien Koch, on behalf of all authors

15

**Editor:**

Comments to the Author:

The reviewer acknowledged the improvement of the manuscript. Still, there are parts needing further revisions; mainly for more precise language and more critical evaluation of used methods and found

20 results.

The manuscript can only be accepted for publication if those are corrected accordingly. Therefore, please carefully go through the general recommendation as well as the very detailed suggestions given in the attachment to further improve the manuscript.

Response: We agree that the referee gave very detailed and helpful comments for improving the

25 manuscript. Hence, we specified the manuscript's language and as suggested critically evaluated the used methods and results for improving the manuscript.

**Dear Referee #2,**

30    we would like to thank you for your time and the constructive comments, which helped to improve the quality of the manuscript. Please find our detailed replies on the comments below. We hope that we answer all your remarks.

In general, our replies to the referee's comments are highlighted in blue. To highlight the nature of our

35    replies we use a traffic light system indicating agreement with the referee marked in green, partial agreement in yellow, and objections in red.

Best regards,

Fabien Koch, on behalf of all authors

40

**Referee #2:**

The MS has been considereably improved in the second round of revision. Well done. However, there is a few issues left that needs to be considered when submitting the final version ready for publication. First, the abstract needs to be polished. I provided some suggestions in the attached pfd version of the MS.

45    Second, I really ask the authors to replace some of the 'non-scientific' phrasing by clear expressions. Here is one example: To my opinion, it is not an appropriate expression to write "the ecological status of groundwater is O.K." but "the ecological status of groundwater was found very good or good applying tier-one of the groundwater ecosystem status index (GESI)". I sugest to replace all 'gw status is O.K.' statements with mor scientific expressions. Third, the authors say that they only applied tier-one of the

50    GESI in their work because their intention was a first evaluattion of gw fauna in Karlsruhe. I am fine with that. However, i the M&M section it is mentioned that, whenever the tier-one assessment leads to a status that is not very godd or good, tier two shuld follow which involved the deliniation of a regional or even local natural reference status. Groundwater ecosystems in Germany are very heterogeneous. As such, it is very likely that the reference values provided by Griebler et al. (2014) are not perfect/ideal for the

55    situation in Karlsruhe. In such as case, considering local or regional peculiarities and the definition of

site-specific reference values would lead to much more reliable results of the assessment. This point needs to be discussed and best also mentioned in the abstract (see my sugestions in the pdf). The important point here is that the authors cannot be sure that applying tier-two would lead to a much better separation of forested sites and residential/commercial/industrial sites. Or in other words, failing of tier-one of the GESI approach may be caused by lack of a local reference sata set. Forth, all sites investigated are in urban area (city of Karlsruhe). I suggest to distinguish the urban area into the two categories 1. residental/commercial/industrial areas and 2. forested areas. A forest in a city is maybe not a 'natural' area. A few more minor points are highlighted in the pdf file.

Response: Thank you for the critical assessment of our study. We address your specific comments in detail below. We fully agree to reformulate parts of the abstract, replace the 'non-scientific' phrasing by clear expressions in the whole manuscript and discuss the issue of local reference values (see 'specific comments').

**Specific comments**

Comment #1. First, the abstract needs to be polished. I provided some suggestions in the attached pfd version of the MS.

Response: We agree to reformulate parts of the abstract. Our changes are listed in the following:

Line 12: We agree to delete the word 'tier'. Done.

Lines 13-14: We agree to replace the word 'assess' by 'examine', because in this context 'explored' sounds to dramatically in our opinion. Moreover, we agree to replace 'condition' by 'status' as well as to delete the words 'the groundwater'. Furthermore, we agree to add 'in the city of' and to delete the last part of the sentence ('and a nearby forest').

Lines 14-16: We do not agree to add 'a' in front of classification, but we agree to add the other suggestions. Thus, we have reformulated the sentence as follows:

"For classification, we apply the groundwater ecosystem status index (GESI), in which a threshold of more than 70 % of Crustaceans and of less than 20 % of Oligochaetes serves as an indication for very good and good ecological conditions."

Lines 16-17: We partially agree to distinguish the urban area into the two categories. We agree that a forest outside a city area might not classify as a 'natural' area. Moreover, parts of the forest containing the measurement wells belong to the districts 'Neureut' and 'Waldstadt' and therefore to the city area of Karlsruhe. Thus, we agree to reformulate the phrasing in the sentence:

"Our study reveals that only 35 % of the wells in the residential, commercial and industrial areas and 50 % of wells in the forested area fulfil these criteria."

Lines 17-18: We agree to delete the beginning of the sentence and 'and nitrate concentrations'.

Lines 19-21: We agree to reformulate the sentence and thus accepted the suggestions:

"Nevertheless, there are noticeable differences in the spatial distribution of species in combination with abiotic groundwater characteristics in groundwater of the different areas of the city, which indicate that a more comprehensive assessment is required to evaluate the groundwater ecological status in more detail."

Lines 21-23: We agree and added the suggested, final sentence together with further information:

"In particular, more indicators, such as groundwater temperature, indicator species, delineation of site-specific characteristics and natural reference conditions should be considered."

Comment #2: Lines 26 & 31: This is not an appropriate journal to be cited if there is alternative publications to be cited.

Response: We agree that there are alternative publications to be cited. Thus, we replaced the study of Avramov et al. (2010) by (German Environment Agency, 2018) in the first sentence of the paragraph and by Griebler and Avramov (2015) and Boulton et al. (2008) in the last sentence (see manuscript).

Comment #3: Line 78-79: Is tthis true. I would rather say, tehre is already some threshold or target values published but none of these have been implemented in official regulations and water law.

Response: We agree and thus added this information as follows:

110 "Until now, there are scientifically derived threshold values for groundwater temperature in the case of thermal (heat) pollution published, but none of these have been implemented in official regulations or water law (Hähnlein et al., 2010, 2013; Blum et al., 2021)."

Comment #4: Line 91ff: The GESI of Griebler et al. (2014) builds on the asessmen scheme of Korbel & Hose 2011. Better to mention this one i front of the GESI approach.

115 Response: We agree. Therefore, we placed the approaches in chronological order.

Comment #5: Lines 112-113: I sugest to distinguish between 1. residential, commercial, and indistrial areas, and 2. forested areas within the 'urban' area of the citay of Karlsruhe.

Response: As already mentioned in Comment #1, we agree and added the suggestions.

120 "The objective of this study is to investigate specifically the groundwater fauna beneath residential, commercial and industrial, i.e. urban areas in comparison to a forested area outside the built-up area of Karlsruhe to determine whether land use has an impact on groundwater faunal communities."

Moreover, we apply this change in the whole manuscript, like for example at the beginning of 125 chapter 2.2 and 3.1.

Comment #6: Line 189-190: This is a weired sentence. to mny repetitions. rephrase!

Response: We agree. Hence, we rephrased the sentence as follows:

"If an ecological assessment of groundwater ecosystems, which is based on groundwater fauna 130 analysis, takes place, some faunistic criteria must be considered."

Comment #7: Line 197: In theory, when tier-one does not deliver a clear result or 'negative' results one need to go to tier-two (Korbel & Hose 2011, Griebler et al. 2014). Here you stop at tier-one, although many wells are classified affected. It is very likely that goig on with tier-two and definig a location-specific reference data set for 'natural conditions' will lead to a new outcome. As such, one can only judge about the two-tiered approach after useing both tiers.

Response: As it is mentioned in the manuscript, our aim was a first screening of an urban area, whereas we only applied Level 1 in our study. We agree, that one can only judge about the Level 2 approach after using it. Therefore, we mentioned in the abstract and conclusion that the delineation of site-specific natural reference conditions and the use of Level 2 will be a next logical step. Moreover, we added information to the discussion of chapter 3.3 as follows:

"Because of heterogeneous groundwater ecosystems in Germany it is likely that reference values provided by Griebler et al. (2014) do not reflect the situation in Karlsruhe correctly. Considering site-specific characteristics and reference values would lead to a more robust assessment."

Comment #8: Line 244: If this is the average nitrate concentration of all wells then it cannot be a range from xx-xy, but one value with a standard deviation, isn't it?

Response: We agree that this sentence can be misunderstood. In this case we mean that the average nitrate contents of all wells varies between 1.3 and 14.7 mg/l. To clarify this, we rephrased the sentence as follows:

"In our study area, the average nitrate contents of all wells vary between 1.3 and 14.7 mg/l."

Comment #9: Line 247: 'most likely caused' instead of 'caused'

Response: We agree and added 'most likely'.

Comment #10: Line 249: 'promotes' instead of 'is characterized by'

Response: We agree and replaced 'is characterized by' by 'promotes'.

Comment #11: Line 269: mention the percentage of colonized wells

Response: We agree. We added the percentage of colonized wells as follows:

"In the pool of samples, 3,666 individuals were detected in 37 of 39 wells, which means that 95 % of the wells are colonised (Table S3).

165

**References**

Avramov, M., Schmidt, S. I., München, C. G., Jürgen, H. and Berkhoff, S.: Dienstleistungen der Grundwasserökosysteme, KW - Korrespondenz Wasserwirtschaft, 3(2), 74–81, doi:10.3243/kwe2010.02.001, 2010.

170 Blum, P., Menberg, K., Koch, F., Benz, S. A., Tissen, C., Hemmerle, H. and Bayer, P.: Is thermal use of groundwater a pollution?, J. Contam. Hydrol., 103791, doi:10.1016/j.jconhyd.2021.103791, 2021.

Boulton, A. J., Fenwick, G. D., Hancock, P. J. and Harvey, M. S.: Biodiversity, functional roles and ecosystem services of groundwater invertebrates, Invertebr. Syst., 22(2), 103–116, doi:10.1071/IS07024, 2008.

175 German Environment Agency: Bericht des Bundesministeriums für Gesundheit und des Umweltbundesamtes an die Verbraucherinnen und Verbraucher über die Qualität von Wasser für den menschlichen Gebrauch* (Trinkwasser) in Deutschland 2014 – 2016, Dessau-Roßlau., 2018.

Griebler, C. and Avramov, M.: Groundwater ecosystem services: A review, Freshw. Sci., 34(1), 355–367, doi:10.1086/679903, 2015.

180 Griebler, C., Stein, H., Hahn, H. J., Steube, C., Kellemrann, C., Fuchs, A., Berkhoff, S. and Brielmann, H.: Entwicklung biologischer Bewertungsmethoden und -kriterien für Grundwasserökosysteme, Umweltbundesamt., 2014.

Hähnlein, S., Bayer, P. and Blum, P.: International legal status of the use of shallow geothermal energy, Renew. Sustain. Energy Rev., 14(9), 2611–2625, doi:10.1016/j.rser.2010.07.069, 2010.

185 Hähnlein, S., Bayer, P., Ferguson, G. and Blum, P.: Sustainability and policy for the thermal use of shallow geothermal energy, Energy Policy, 59, 914–925, doi:10.1016/j.enpol.2013.04.040, 2013.